# Single-cell growth inference of *Corynebacterium glutamicum* reveals asymptotically linear growth

**Joris JB Messelink[1†], Fabian Meyer[2,3†], Marc Bramkamp[2,3]\*, Chase P Broedersz[1,4]\***

[1]Arnold-Sommerfeld-Center for Theoretical Physics, Ludwig-Maximilians-Universität München, Munich, Germany; [2]Ludwig-Maximilians-Universität München, Fakultät Biologie, Planegg-Martinsried, Germany; [3]Christian-Albrechts-Universität zu Kiel, Institut für allgemeine Mikrobiologie, Kiel, Germany; [4]Department of Physics and Astronomy, Vrije Universiteit Amsterdam, Amsterdam, Netherlands

**Abstract** Regulation of growth and cell size is crucial for the optimization of bacterial cellular function. So far, single bacterial cells have been found to grow predominantly exponentially, which implies the need for tight regulation to maintain cell size homeostasis. Here, we characterize the growth behavior of the apically growing bacterium *Corynebacterium glutamicum* using a novel broadly applicable inference method for single-cell growth dynamics. Using this approach, we find that *C. glutamicum* exhibits asymptotically linear single-cell growth. To explain this growth mode, we model elongation as being rate-limited by the apical growth mechanism. Our model accurately reproduces the inferred cell growth dynamics and is validated with elongation measurements on a transglycosylase deficient Δ*rodA* mutant. Finally, with simulations we show that the distribution of cell lengths is narrower for linear than exponential growth, suggesting that this asymptotically linear growth mode can act as a substitute for tight division length and division symmetry regulation.

**\*For correspondence:**
bramkamp@ifam.uni-kiel.de (MB);
c.p.broedersz@vu.nl (CPB)

[†]These authors contributed equally to this work

**Competing interests:** The authors declare that no competing interests exist.

## Introduction

Regulated single-cell growth is crucial for the survival of a bacterial population. At the population level, fundamental laws of growth were discussed as early as the beginning of the 20th century, and distinct population growth phases were identified and attributed to bacterial growth (*Lane-Claypon, 1909*; *Buchanan, 1918*; *Monod, 1949*). At the time, however, growth behavior at the single-cell level remained elusive. This changed only over the last decade, as evolving technologies enabled detailed measurements of single-cell growth dynamics. Extensive work was done on common model organisms, including *Escherichia coli*, *Bacillus subtilis,* and *Caulobacter crescentus,* revealing that averaged over the cell cycle, single cells grow exponentially for these species (*Taheri-Araghi et al., 2015*; *Mir et al., 2011*; *Iyer-Biswas et al., 2014*; *Yu et al., 2017*; *Godin et al., 2010*).

Single-cell exponential growth is expected if cellular volume production is proportional to the protein content (*Amir, 2014*), as shown to be the case for *E. coli* (*Belliveau et al., 2020*). Importantly, however, such a proportionality will only be present if cellular volume production is rate-limiting for growth. Cells with different rate-limiting steps could display distinct growth behavior. Recently, detailed analysis of the mean growth rate throughout the cell cycle revealed deviations from pure exponential growth. For *B. subtilis* (*Nordholt et al., 2020*), a biphasic growth mode was observed, where a phase of approximately constant elongation rate is followed by a phase of increasing elongation rate. For *E. coli,* a new method provides evidence for super-exponential in the later stages of the cell cycle (*Kar et al., 2021*).

A promising candidate for uncovering strong deviations from exponential growth is the Gram-positive *Corynebacterium glutamicum*. This rod-shaped bacterium grows its cell wall exclusively at the cell poles, allowing, in principle, for deviations from exponential single-cell growth (*Figure 1*). The dominant growth mode depends on the rate-limiting step for growth, which is presently unknown for this bacterium. Non-exponential growth modes may have important implications for growth regulatory mechanisms: while exponential growth requires checkpoints and regulatory systems to maintain a constant size distribution (*Mir et al., 2011*), such tight regulation might not be needed for other growth modes.

*Corynebacterium glutamicum* is broadly used as a production-organism for amino-acids and vitamins and also serves as model organism for the taxonomically related human pathogens *Corynebacterium diphtheriae* and *Mycobacterium tuberculosis* (*Hermann, 2003*; *Antoine et al., 1988*; *Schubert et al., 2017*). A common feature of Corynebacteria and Mycobacteria is the existence of a complex cell envelope. The cell wall of these bacteria is a polymer assembly composed of a classical bacterial peptidoglycan (PG) sacculus that is covalently bound to an arabinogalactan (AG) layer (*Alderwick et al., 2015*). Mycolic acids are fused to the arabinose and form an outer membrane like

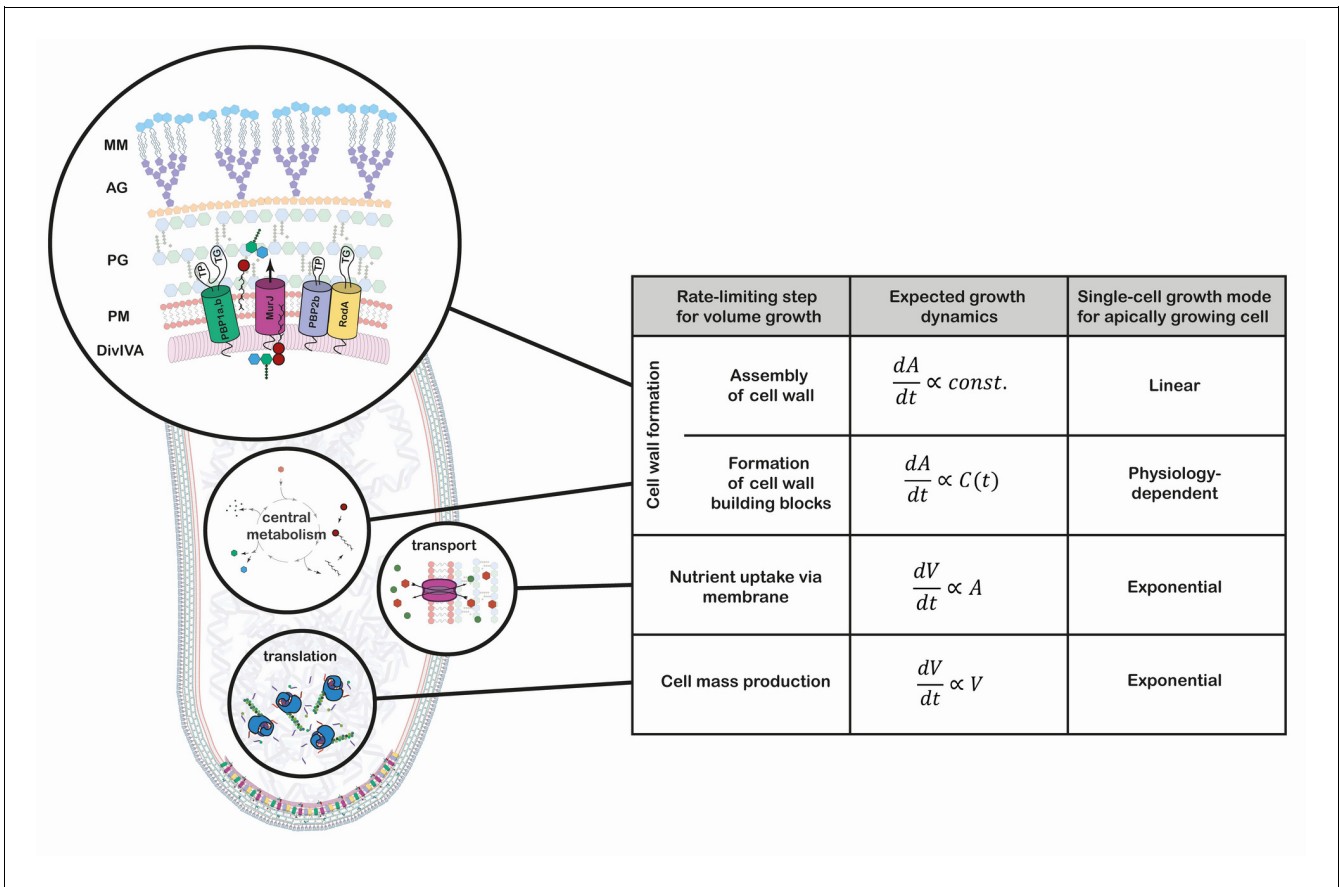

| Rate-limiting step for volume growth | Expected growth dynamics | Single-cell growth mode for apically growing cell |
|---|---|---|
| **Assembly of cell wall** | $\frac{dA}{dt} \propto const.$ | Linear |
| **Formation of cell wall building blocks** | $\frac{dA}{dt} \propto C(t)$ | Physiology-dependent |
| **Nutrient uptake via membrane** | $\frac{dV}{dt} \propto A$ | Exponential |
| **Cell mass production** | $\frac{dV}{dt} \propto V$ | Exponential |

**Figure 1.** Growth mode analysis for four possible rate-limiting steps for cellular volumegrowth in the apically growing *C. glutamicum*. Here, $V$ is the cellular volume, $A$ is the cell wall area, and $C(t)$ is the concentration of membrane building blocks in the cytoplasm. A constant cell width is assumed throughout, implying $\frac{dA}{dt} \propto \frac{dV}{dt}$. A fixed production capacity per unit volume is assumed for the rate-limiting steps 'cell mass production' and 'formation of cell wall building blocks'. For the rate-limiting step 'assembly of cell wall', a constant insertion area at the cell poles is assumed. For an analysis of the single-cell growth mode if cell wall building block formation is the rate-limiting step for growth, see Appendix 1. Cell mass production, specifically ribosome synthesis, has previously been indicated as the rate-limiting step for growth in *E. coli* (*Belliveau et al., 2020*; *Scott et al., 2010*; *Amir, 2014*). Linear growth is observed if the rate-limiting step for volume growth is the cell wall assembly (shown here in a simplified representation). The protein DivIVA serves as a scaffold at the curved membrane of the cell pole for the recruitment of the Lipid-II flippase MurJ and several mono- and bi-functional trans-peptidases (TP) and -gylcosylases (TG). In the process of elongation, peptidoglycan (PG) precursors are integrated into the existing PG sacculus, which serves as a scaffold of the synthesis of the arabinogalactan-layer (AG) and the mycolic-acid bilayer (MM).

bilayer, rendering the cell surface highly hydrophobic (*Puech et al., 2001*). The mycolic acid membrane (MM) is an efficient barrier that protects the cells from many conventional antibiotics.

*C. glutamicum's* growth and division behavior is vastly different to that of classical model species. In contrast to rod-shaped firmicutes and γ-proteobacteria, where cell-wall synthesis is dependent on the laterally acting MreB, members of the *Corynebacterianeae* lack a *mreB* homologue and elongate apically. This apical elongation is mediated by the protein DivIVA, which accumulates at the cell poles and serves as a scaffold for the organization of the elongasome complex (*Letek et al., 2008*; *Hett and Rubin, 2008*; *Sieger et al., 2013*; *Figures 1* and *2A,B*). Furthermore, a tightly regulated division-site selection mechanism is absent in this species. Without harboring any known functional homologues of the Min- and nucleoid occlusion (Noc) system, division typically results in unequally sized daughter cells (*Donovan et al., 2013*; *Donovan and Bramkamp, 2014*). Lastly, the spread in growth times between birth and division is much wider than in other model organisms, suggesting a weaker regulation of this growth feature (*Donovan et al., 2013*). These atypical growth properties suggest that this bacterium is an interesting candidate to search for novel growth modes. To reveal the underlying growth regulation mechanisms, it is necessary to study the elongation dynamics of *C. glutamicum*.

Here, we measure the single-cell elongations within a proliferating population of *C. glutamicum* cells, and develop an analysis procedure to infer their growth behavior. We find that *C. glutamicum* deviates from the generally assumed single-cell exponential growth law. Instead, these *Corynebacteria* exhibit asymptotically linear growth. We develop a mechanistic model, termed the rate-limiting apical growth (RAG) model, showing that these anomalous elongation dynamics are consistent with the polar cell wall synthesis being the rate-limiting step for growth. Finally, we demonstrate a connection between mode of growth and the impact of single-cell variability on the cell size distribution of the population. For an asymptotically linear grower, these variations have a much smaller impact on this distribution than they would for an exponential grower, which may suggest an evolutionary explanation for the lack of tight regulation of single-cell growth in *C. glutamicum*.

## Results

### Measuring elongation trajectories using microfluidic experiments

To measure the development of single *C. glutamicum* cells over time, we established a workflow combining single-cell epifluorescence microscopy with semi-automatic image processing. Cells were grown in a microfluidic device. We used wild type cells and cells expressing the scaffold protein DivIVA as a translational fusion to mCherry. DivIVA is used as a marker for cell cycle progression, since it localizes to the cell poles and to the newly formed division septum in *C. glutamicum* (*Letek et al., 2008*; *Donovan et al., 2013*).

For the choice of microfluidic device, we deviate from the commonly used Mother Machine (*Long et al., 2013*), which grows bacteria in thin channels roughly equaling the cell width. The Mother Machine is not ideally suited for *C. glutamicum* growth, as the characteristic V-snapping at division could lead to shear forces and stress during cell separation, affecting growth (*Zhou et al., 2019*). Indeed, in some cases, the mother machine has been shown to affect growth properties even in cells not exhibiting V-snapping at division, due to mechanical stresses inducing cell deformation (*Yang et al., 2018*). Therefore, we instead used microfluidic chambers that allow the growing colony to expand without spatial limitations into two dimensions for several generations (*Figure 2C,D*, Materials and methods). Within the highly controlled environment of the microfluidic device, a steady medium feed and a constant temperature of 30°C was maintained. We extracted bright-field- and fluorescent-images over 3-min intervals, which were subsequently processed semi-automatically with a workflow developed in FIJI and R (*Schindelin et al., 2012*; *R Development Core Team, 2003*). For each individual cell per time-frame, the data set contains the cell's length, area and estimated volume, the DivIVA-mCherry intensity profile, and information about generational lineage (*Figure 2E–G*). We used these data sets to further investigate the growth behavior of our bacterium. Thus, using this procedure, we obtained data sets containing detailed statistics on single-cell growth of *C. glutamicum*.

For subsequent analysis, the measured cell lengths were used, because of their low noise levels as compared to other measures (*Appendix 2—figure 1B*). Importantly, the increases in cell length

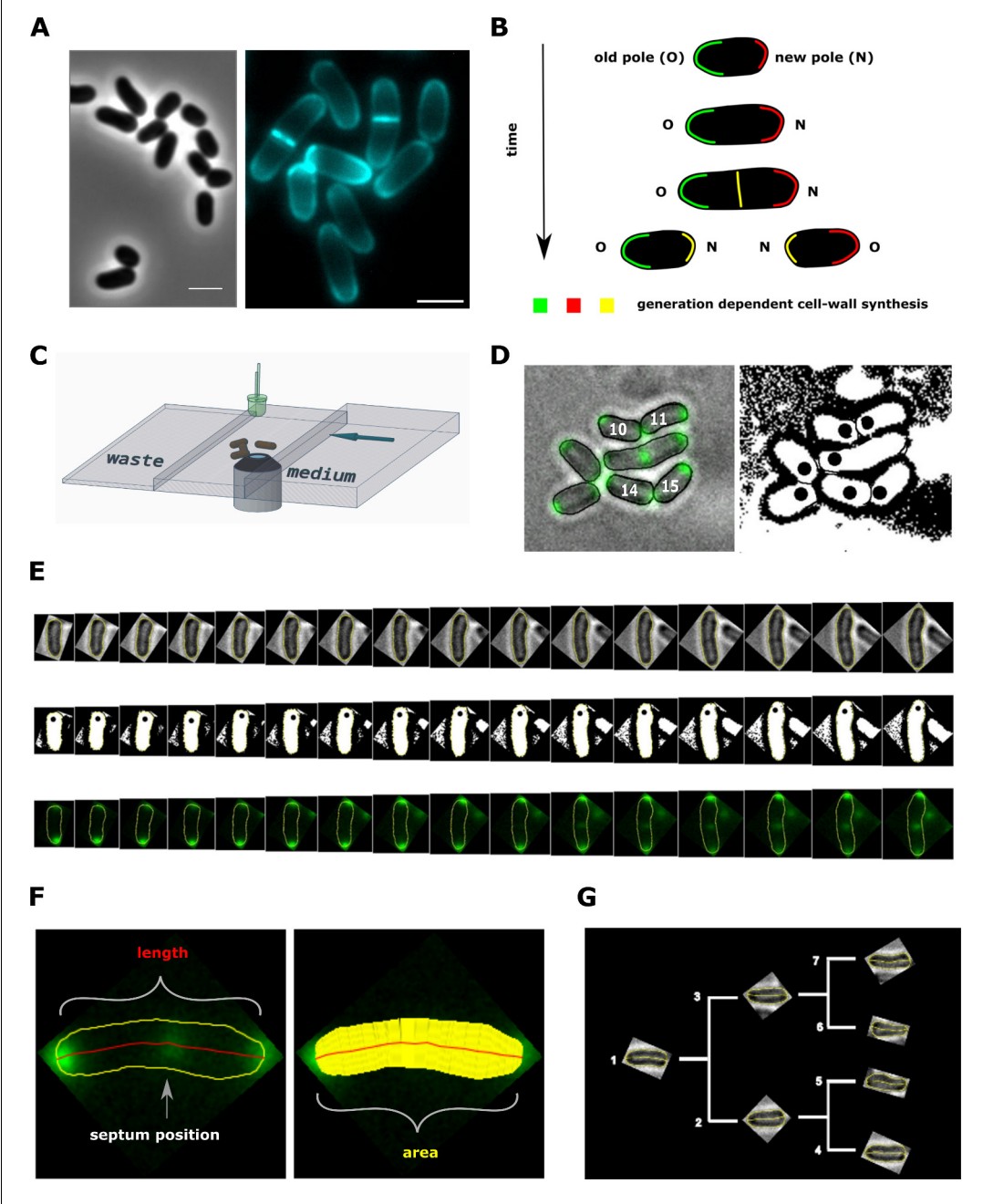

**Figure 2.** Experimental procedure and image analysis. (**A**, left) Phase contrast image of *C. glutamicum* in logarithmic growth phase, indicating the variable size of daughter cells. (**A**, right) HADA labeling of nascent peptidoglycan (PG), indicating the asymmetric apical growth where the old cell-pole always shows a larger area covered compared to the new pole. The labeling also reveals the variable septum positioning; Scale bar: 2 μm (**B**) Schematic showing the generation-dependent sites of PG synthesis in *C. glutamicum*, including the maturation of a new to an old cell-pole. (**C**) Illustration of the microfluidic device for microscopic monitoring of a growing colony. (**D**) Example screen-shot of the developed method to extract individual cell cycles from a multi-channel time-lapse micrograph. The left panel shows a merging of the bright-field channel and the mCherry-tagged DivIVA together with an individual ID# that is assigned to cells right after division. The black dots in the right panel indicate the new cell pole. (**E**) Example of an extracted individual cell cycle from birth (left) until prior to division (right), showing the bright-field (top), the orientation (middle) and the localization of mCherry-tagged DivIVA (bottom). (**F**) Example of the developed single cell analysis algorithm, measuring the length according to the cell's geometry, as well as the cell's area and the septum position relative to the new pole. (**G**) Dendrogram providing the rationale for identification of single cells in a growing colony.

are proportional to the increases in cell area (*Appendix 2—figure 1A*), suggesting that cellular length increase is also proportional to the volume increase. This proportionality is expected since the rod-shaped *C. glutamicum* cells insert new cell wall material exclusively at the poles, while maintaining a roughly constant cell width over the cell cycle (*Schubert et al., 2017*; *Daniel and Errington, 2003*).

## Population-average test suggests non-exponential growth for *C. glutamicum*

A standard way of characterizing single-cell bacterial growth, is to determine the average relation between birth length $l_b$ and division length $l_d$ (*Amir, 2014*). For *C. glutamicum*, we find an approximately linear relationship between these birth and division lengths, with a slope of 0.91±0.16 (2xSEM, *Figure 3A*). This indicates that on a population level, *C. glutamicum* behaves close to the *adder* model, in which cells on average grow by adding a fixed length before dividing (*Jun and Taheri-Araghi, 2015*; *Amir, 2014*).

To investigate the growth dynamics from birth to division, we first tested if our cells conform to the generally observed exponential mode of single-cell growth. To this end, we applied a previously developed analysis on bacterial elongation data (*Logsdon et al., 2017*), by plotting $\ln\left(\frac{l_d}{l_b}\right)$ versus the growth time (*Figure 3B*). For an exponential grower, with the same exponential growth rate $\alpha$ for all cells, the averages of $\ln\left(\frac{l_d}{l_b}\right)$ per growth time bin are expected to lie along a straight line with slope $\alpha$ intersecting the origin. By contrast, there appears to be a systematic deviation from this trend, with cells with shorter growth times lying above this line and cells with longer growth times lying below it, suggesting non-exponential elongation behavior. However, the significance and implications of these deviations for single-cell growth behavior are not clear from this analysis. There are several quantities that could be highly variable between cells that are averaged out in this representation, such as possible variations in exponential growth rate as a function of birth length, or variations in growth mode over time. Furthermore, it was recently shown that exponentially growing cells can appear non-exponential with this test in the presence of noise in the exponential growth rate (*Kar et al., 2021*). Thus, a more detailed analysis of the growth trajectories is needed to rule out exponential growth, and to quantitatively characterize the growth dynamics.

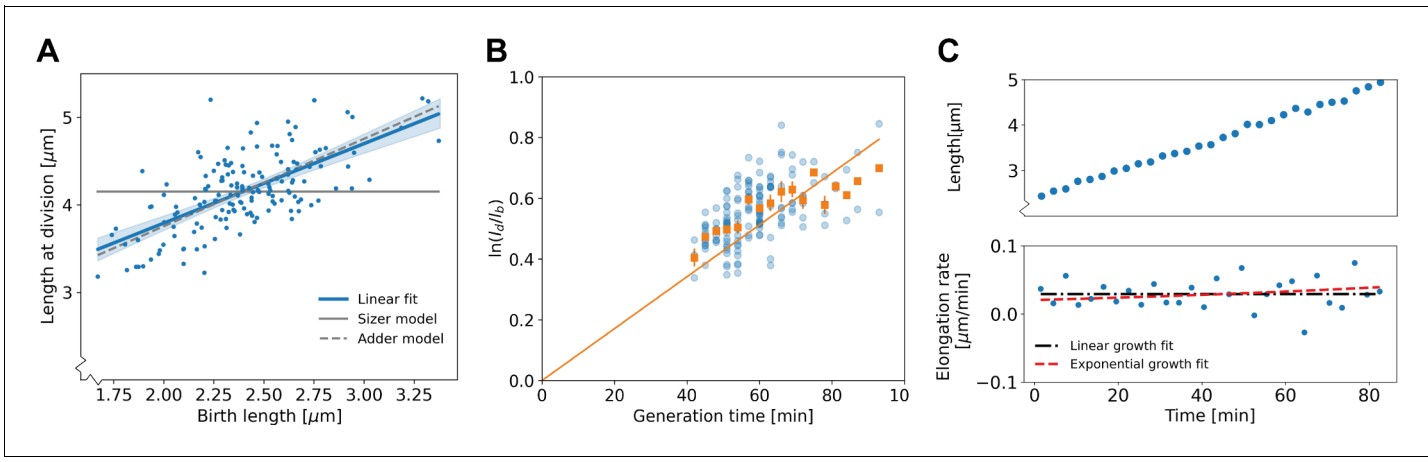

**Figure 3.** Population-level and single-cell level growth analysis. (A) Birth length $l_b$ plotted against division length $l_d$ for all measured cells, together with a linear fit (blue line), which has a slope of 0.91±0.16. Gray solid line: best fit assuming a pure sizer (slope 0). Gray dashed line: best fit assuming a pure adder (slope 1). The 95% confidence intervals of the linear fit, obtained via bootstrapping, are indicated by the blue shaded region. (B) Generation time versus $\ln\left(\frac{l_d}{l_b}\right)$ for all cells (blue dots) and the average per generation time (orange squares), with the standard error of the mean shown for all generation times for which at least three data points are available. The orange line represents a linear fit through the generation time averages that passes through the origin. For exponential growth, the averages would lie along this line, and the slope would be equal to the exponential growth rate. (C) Growth trajectory for a single cell (upper panel), together with its derivative for each measurement interval (lower panel). Fits to the derivative are shown for linear growth (black dash-dotted line) and exponential growth (red dashed line).

The variability of key growth parameters is not easily extracted from individual growth trajectories due to the inherent stochasticity of the elongation dynamics and measurement noise (*Figure 3C*). In fact, it has been estimated that to distinguish between exponential and linear growth for an individual trajectory, the trajectory needs to be determined with an error of ~6% (*Cooper, 1998*). Distinguishing subtler growth features may require an even higher degree of accuracy, which is presently experimentally unavailable (Appendix 3). Therefore, an analysis method is needed that is less noise-sensitive than an inspection of the single-cell trajectories, but simultaneously does not average out potentially relevant growth features such as time-dependence and birth length variability.

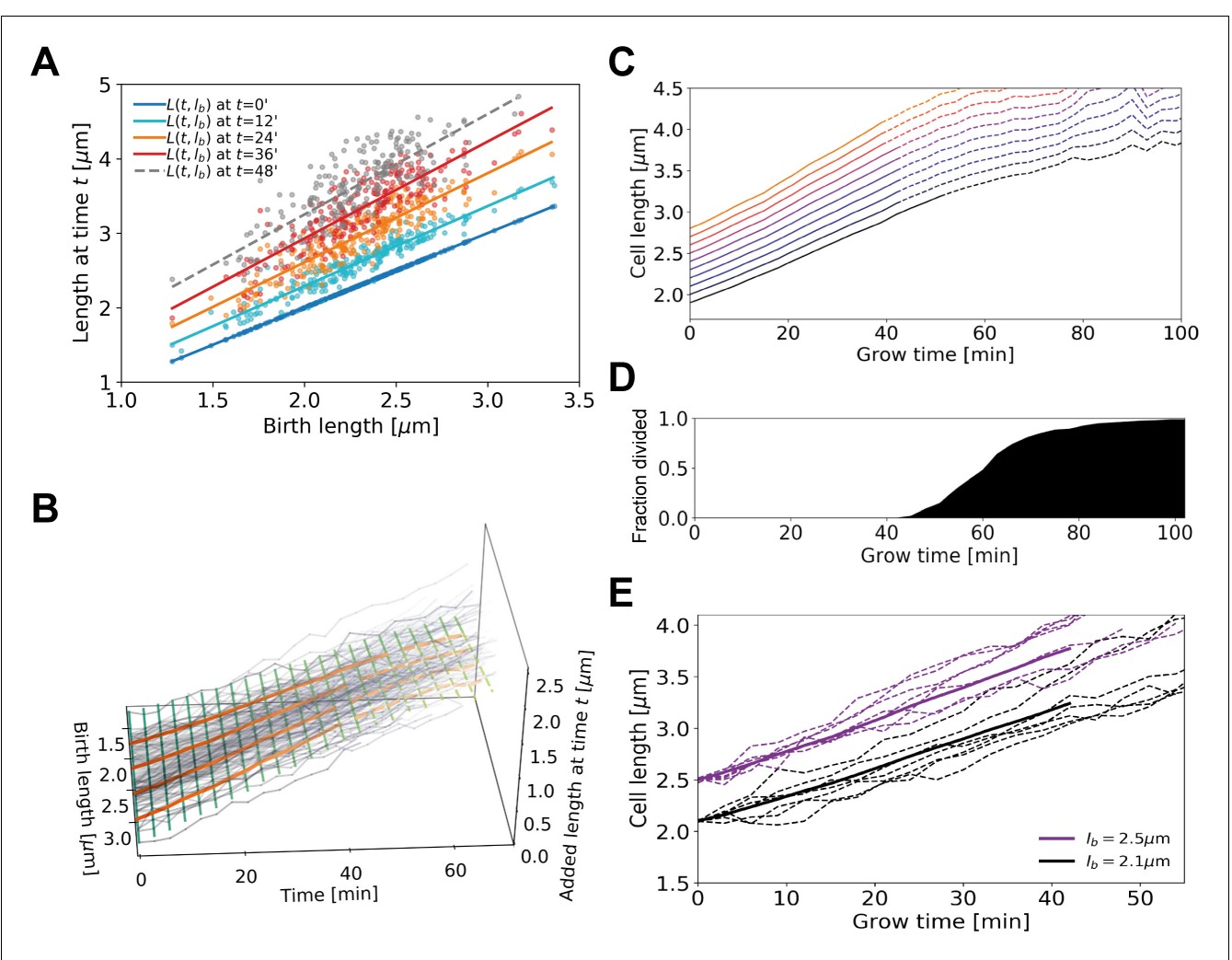

**Figure 4.** Average elongation curve inference procedure. (**A**) For each cell, the length $L(t)$ at different times $t$ since birth is plotted as a function of birth length $l_b$. A linear fit of the resulting 'wave front' is performed for each time $t$. This allows us to determine average cell length $L(t, l_b)$ at time $t$ as a function of birth length $l_b$. (**B**) 3D representation of the inference method of average length trajectories, with the added length $L(t, l_b) - l_b$ on the z-axis. Elongation trajectories for individual cells are indicated in gray, linear fits through all cell lengths at each timestamp are indicated by green lines. The orange lines represent four sample average length trajectories, obtained by connecting all values of the green lines associated with one birth length. Dotted lines represent regimes where averages are biased due to dividing cells. (**C**) Average elongation trajectories obtained from the fits shown in (**A**) for a range of birth lengths, starting at 1.9 μm with steps of 0.1 μm (solid lines). The dashed lines represent regions where the inferred elongation curves are biased due to dividing cells, and are excluded from subsequent analysis. (**D**) Cumulative fraction of cells divided as a function of grow time. (**E**) Elongation trajectories for cells with birth lengths close to 2.5 μm (purple dashed lines) and birth lengths close to 2.1 μm (black dashed lines) together with their respective inferred average trajectories (purple solid line and black solid line).

## Growth-inference method yields average elongation rate curves

To obtain quantitative elongation rate curves as a function of time and birth length, despite the high degree of individual variation, we developed a data analysis procedure that exploits the noise-reducing properties of multiple-cell conditional averaging. The key idea is to obtain an average dependence of the cellular length $L(t, l_b)$ on the time $t$ since birth and birth length $l_b$, by first obtaining the average dependence of $L(t, l_b)$ on $l_b$ for each discrete value of $t$ individually. This yields an average elongation curve for each birth length $l_b$, without the need to perform inference on noisy $L(t)$ single-cell curves.

The analysis procedure is as follows. First, for all cells in our data set, we determine the time since birth $t$, the cellular length $L$ at time $t$, and the birth length $l_b$. Subsequently, we relate the length at time $t$ to the birth length, yielding a series of scatter plots for each measurement time (*Figure 4A*). Importantly, these scatterplots suggest a simple apparently linear relationship between $L$ and $l_b$. For each such plot, we thus make a linear fit through the data, yielding a family of curves for each time since birth $t$ (*Figure 4B*). Higher-order fitting functions result in a negligible improvement of the goodness-of-fit, while increasing the mean error on inferred elongation rates (*Appendix 2—figure 3*). Note that for both purely linear and purely exponential growth, would depend linearly on: for linear growth $L(t, l_b) = \alpha t + l_b$, whereas for exponential growth $L(t, l_b) = l_b \exp(\alpha t)$ (*Appendix 2—figure 3*). From the family of relations, we compute a series of points $\{L(t_0, l_b), L(t_1, l_b), L(t_2, l_b)\}$ yielding the average growth trajectory of a cell starting out at length $l_b$ (*Figure 4C*). Note, we must remove a bias in the $l_b$ associated with each average trajectory, arising from measurement noise in the cell lengths at birth (Appendix 4). In summary, this procedure allows us to obtain an unbiased interference of the average elongation trajectories as a function of the cell's birth length.

## Elongation rate inference reveals asymptotically linear growth mode

Our inference approach yields the functional dependence of the average added length on growth time and birth length. We find that the average length steadily increases initially, but levels off and shows pronounced fluctuations for larger growth times (*Figure 4C*). This late-time behavior (dashed lines in *Figure 4C*) is caused by decreasing cell numbers due to division events (*Figure 4D*), which also introduces a bias in the averaging procedure. After the first division event, the average inferred growth would be conditioned on the cells that have not divided yet. For a given birth length, faster-

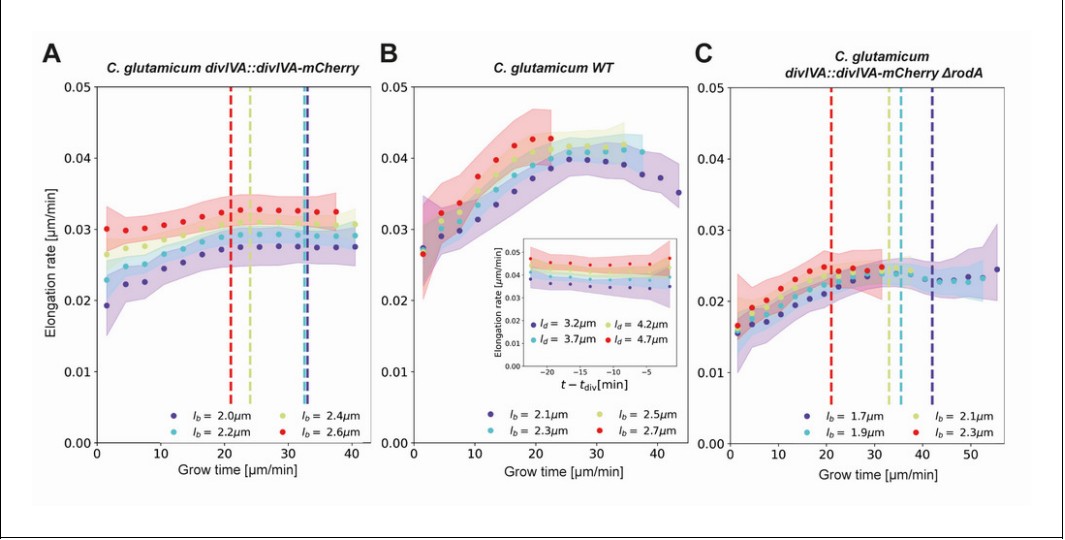

**Figure 5.** Inferred average elongation rates. (A) Average elongation rates for four birth lengths (dots), for the DivIVA-labeled cells. The $2\sigma$ confidence intervals obtained by bootstrapping are indicated by the shaded areas. Vertical dashed lines: average onset of septum formation per birth length. (B) Average elongation rate trajectories for the wild-type cells, confidence intervals shown as in (A). Inset: average elongation trajectories as a function of the time until division. (C) Average elongation rate trajectories for the $\Delta rodA$ mutant, confidence intervals shown as in (A).

growing cells divide earlier than slower-growing cells (*Appendix 2—figure 2*) causing this conditional average to underestimate cellular elongation rates for the whole population after the first division. Because our aim is to infer elongation curves that characterize the whole population, ranging from slow to fast growers, for further analysis only the part of each trajectory before the first division event is used (*Figure 4D*). Sub-population elongation curves can also be obtained that extend past the first division event, but only if the entire analysis for these curves is performed only on these slower-dividing cells (*Appendix 2—figure 4*).

We obtain elongation rate curves by taking a numerical derivative of smoothed growth trajectories (Appendix 5). To determine the associated error margins of the elongation rates, we use a custom bootstrapping algorithm (*Efron, 1979*). The resulting 2σ bounds are shown as semitransparent bands. Despite the high noise level of individual elongation trajectories, the inferred average elongation rates have an error margin of around 8%. Thus, our approach robustly infers average elongation trajectories from single-cell growth data. Elongation rates of cells with larger birth length are consistently higher than the elongation rates of cells with smaller birth length. Strikingly, the elongation rate curves initially increase, but then gradually level off toward a linear growth mode (*Figure 5*). We note a slight difference in the cell elongation rates between the strain expressing DivIVA-mCherry (*Figure 5A*) and wild type cells (*Figure 5B*). Importantly, this difference does not qualitatively change the mode of growth, but does show that a translational fusion to DivIVA tends to lower elongation rates. This likely reflects a disturbance in the interaction between RodA or bifunctional PBPs and the DivIVA-mCherry fusion protein, indicating that the DivIVA-mCherry fusion is not fully functional. This is consistent with findings we reported earlier (*Donovan et al., 2013*).

To further test if the linear growth mode persists until division, we adapt our inference procedure to obtain average elongation curves $L(t - t_d, l_d)$ as a function of the time until division $t - t_d$ and division length $l_d$. The construction is analogous to that of $L(t, l_b)$ (Appendix 6). Calculating the corresponding elongation rate curves, we find that that linear growth indeed extends until the division time across division lengths (inset *Figure 5B,SI*, *Appendix 6—figure 1*). Note that with this construction, elongation rates become biased once $|t - t_d|$ exceeds the shortest single-cell total growth time. Hence, for our analysis we only consider elongation rate curves until this point.

To test the performance of our proposed inference method, we simulated a population of growing cells with a presumed growth mode from which we sample cells lengths as in our experiments, including measurement noise (Appendix 3). We ran simulations for cells performing linear growth, exponential growth, and the growth mode inferred here for DivIVA-labeled cells (*Figure 5A*). We find that our inference method is able to recover the input growth mode with high precision in all cases (Appendix 4, Appendix 7), demonstrating the accuracy and internal consistency of our inference method.

## Onset of the linear growth regime does not consistently coincide with septum formation

A central feature of the obtained elongation rate curves is a transition from an accelerating to a linear growth mode after approximately 20–25 min (*Figure 5*). One possibility is that this levelling off is connected with the onset of division septum formation. Given that the FtsZ-dependent divisome propagates the invagination of the septum under the consumption of cell wall precursors (e.g. Lipid-II), we hypothesized that the appearance of the additional sink for cell-wall building blocks could lead to coincidental leveling-off of the elongation rates (*Scheffers and Tol, 2015*). To test this hypothesis, we used the moment of a sharp increase in the average DivIVA-mCherry signal at the cell center as a proxy for the moment of onset of septum formation (*Appendix 2—figure 7*): the inward growing septum introduces a negative curvature of the plasma membrane, leading to the accumulation of DivIVA (*Lenarcic et al., 2009*; *Strahl and Hamoen, 2012*). We observe that the onset of septum formation does not consistently coincide with the moment at which the elongation rate levels off (*Figure 5A*): for smaller cells, the onset of septum formation occurs much later. Therefore, it seems implausible that the observed linear growth regime is due the septum acting as a sink for cell-wall building blocks.

## Polar cell wall formation is the rate-limiting step for growth, leading to a linear growth regime

To provide insight into the anomalous single-cell growth behavior, we model single-cell elongation as being rate-limited by the apical cell wall formation mechanism. To formulate this rate-limiting apical growth (RAG) model, we first consider the biochemical pathway that leads to cell wall formation in *C. glutamicum*, as illustrated in *Figure 1*. The key process for cell wall formation in *C. glutamicum* is polar peptidoglycan (PG) synthesis. PG intermediates are provided by the substrate Lipid-II, and the integration of new material into the PG-mesh is mediated by transglycosylases (TGs) located at the cell pole. At the TG sites, Lipid-II is translocated across the plasma membrane by the Lipid-II flippase MurJ (*Sham et al., 2014*; *Kuk et al., 2017*; *Butler et al., 2013*). After PG building blocks provided by Lipid-II are incorporated into the existing cell wall by transglycolylation, transpeptidases (TP) conduct the crosslinking of peptide subunits, which contributes to the rigidity of the cell wall (*Scheffers and Pinho, 2005*; *Valbuena et al., 2007*; *Schleifer and Kandler, 1972*). During growth, the area of the PG sacculus, and thus the number of TG sites, is extended by RodA and bifunctional penicillin binding proteins (PBPs), recruited by DivIVA (*Letek et al., 2008*; *Sieger et al., 2013*).

To model this growth mechanism, we assume that the rate of new cell wall formation is proportional to the number of TG sites. We describe the interaction between Lipid-II and TG sites by Michaelis-Menten kinetics (*Figure 6A*). Specifically, if the cell length added per unit time is proportional to the cell wall area added per unit time, we find

$$\frac{dL(t)}{dt} = \alpha \frac{C(t)N(t)}{K_m + C(t)} \tag{1}$$

with $L(t)$ the cell length at time $t$, $C(t)$ the concentration of Lipid-II, $K_m$ the Michaelis constant for this reaction, and $\alpha$ is a proportionality constant.

To gain insight into the cell-cycle-dependence of $N(t)$ and $C(t)$, we made use of the cyan fluorescent D-alanine analogue HADA (see Materials and methods) to stain newly inserted peptidoglycan. Exponentially growing *C. glutamicum* cells were labeled with HADA for 5 min before imaging. The HADA stain will mainly appear at sites of nascent PG synthesis. As expected, HADA staining resulted in a bright cyan fluorescent signal at the cell poles and at the site of septation. Still images were obtained with fluorescence microscopy and subjected to image analysis (*Figures 2A* and *6B*, Materials and methods).

We first verify that the HADA intensity profile at the cell poles can be used as a measure for the peptidoglycan insertion rate. To do this, we assume that the HADA intensity profile has two relevant contributions: fluorescent probe present in the cell plasma, and fluorescent probe attached to newly inserted peptidoglycan. We use the minimum of the HADA intensity profile, consistently located around mid-cell in non-dividing cells, as an estimate of the contribution from the cell plasma in each cell, and subtract this from the entire cellular profile to obtain the corrected HADA profile (*Appendix 2—figure 8*). We then define the polar regions where we use the corrected HADA intensity to measure newly inserted peptidoglycan as the portions of the cell within 0.78 µm of the cell tips. Our results are, however, not strongly dependent on this polar region definition (*Appendix 2—figure 10*). Subsequently, we compute a moving average of the corrected polar HADA intensity as a function of cell length (*Figure 6C*). These polar HADA intensities are approximately proportional to the inferred average single-cell elongation rates (Appendix 8), as shown in the inset of *Figure 6C*. Thus, the polar HADA intensities can be used as a measure for the cellular elongation rate. Assuming a proportional relationship between elongation rate and peptidoglycan insertion rate, this implies the polar HADA intensities are also approximately proportional to the peptidoglycan insertion rate. Deviations of ~10% from proportionality within the error margins observed over the range of tip intensities do not affect subsequent conclusions from the HADA intensity data.

Analyzing the HADA intensity profile for smaller segments within the polar region, we find that the increase in intensity is unevenly distributed (*Figure 6D*). Close to the cell tip, the HADA intensity remains approximately constant across cell lengths, whereas a linear increase over cell lengths is seen further from the tip. Considering the implications of these measured intensities for $C(t)$ and $N(t)$ within our model in *Equation (1)*, we argue for a scenario where either $C(t)$ is constant or $C(t) \gg K_m$. Our reasoning is as follows. From *Equation (1)*, we see that the approximately constant intensity at the cell tip can be produced in two ways: (1) $C(t) \gg K_m$ or $C(t)$ is constant across cell

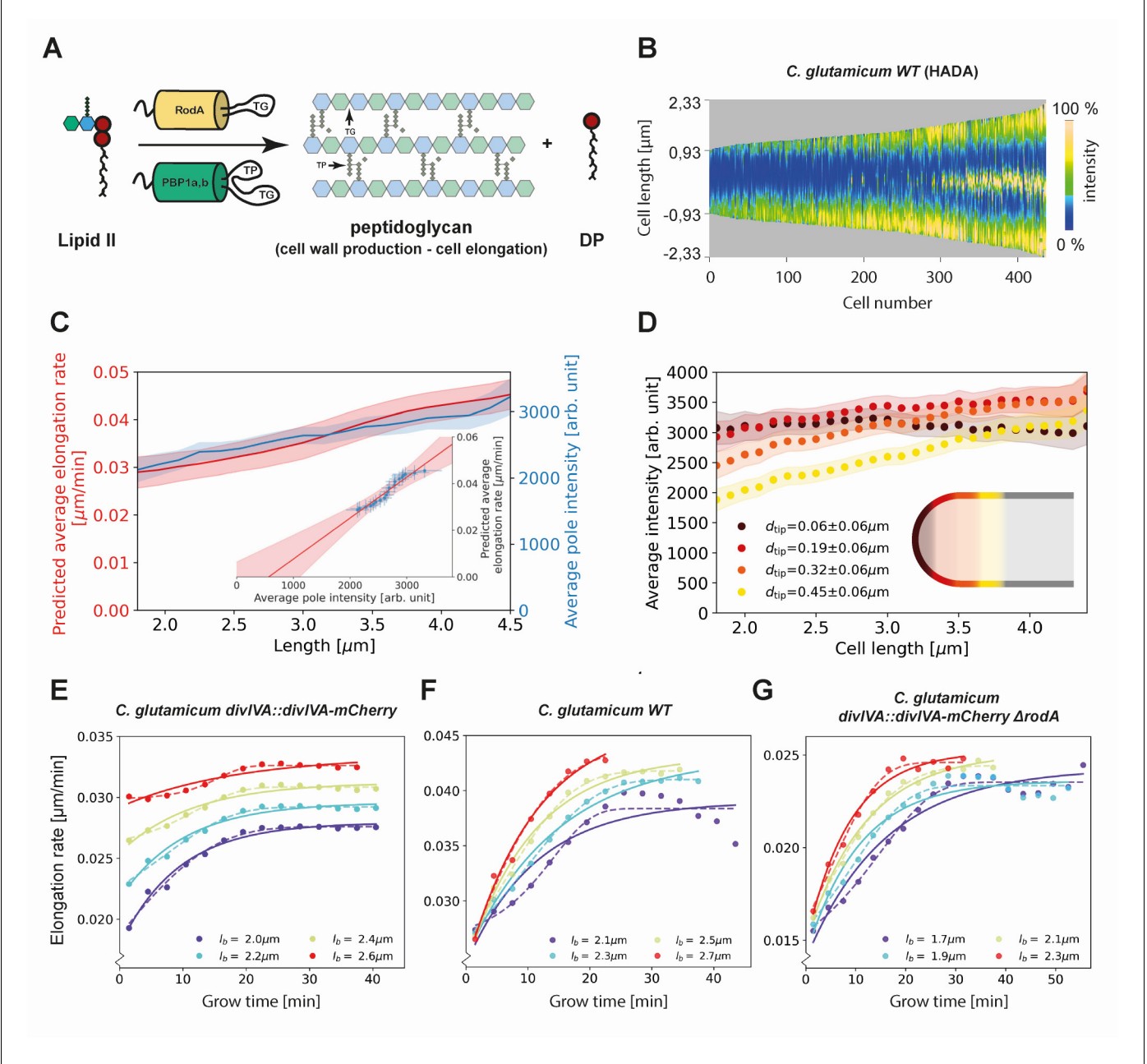

**Figure 6.** Modeling of average elongation rates using HADA staining results. (**A**) Schematic depicting cell wall formation via Lipid-II and transgrlycosylases (TG's). The corresponding Michaelis-Menten equation describes the change of length over time as function of the Lipid-II concentration $C(t)$ and the number of the TG sites $N(t)$. (**B**) Demograph of *C. glutamicum* cells stained with HADA. Cell are ordered by length, with the stronger signal oriented downwards. (**C**) Average elongation rate as a function of cell length (red), predicted from obtained average elongation rate curves (Appendix 8), together with the average HADA staining intensity at the cell pole after background correction (blue). The cell pole is defined here as the region within 0.77 μm (60 pixels) of the cell tip. The shaded regions indicate the 2XSEM bounds. For both curves, a moving average over cells within 0.7 μm of each x-coordinate is applied over the underlying data. Inset: predicted average elongation rate versus average HADA staining intensity (blue dots). A linear fit through the result (red line) is consistent with a proportional relationship. (**D**) Average HADA intensity as a function of cell length, shown for four regions close to the cell tip. A moving average over cells within 0.7 μm of each x-coordinate is applied over the underlying data. (**E-G**) Dots: average elongation rate curves as shown in *Figure 5A*. Solid lines: best fit of elongation model from *Equation (2)*, which assumes constant transglycosylase recruitment. Dashed lines: best fit of elongation model from *Equation (3)*, which assumes an exponential increase of transglycosylase recruitment.

lengths, and the number of transglycosylases at the tip $N_{\text{tip}}(t)$ is constant, or (2) $N_{\text{tip}}(t)$ and $C(t)$ anti-correlate in such a way to produce constant insertion.

However, we consider constant $N_{\text{tip}}(t)$ as biologically the most plausible scenario. This is supported by noting that the concentration of Lipid-II is the same directly before and after division, such that $C(t)$, and by implication $N_{\text{tip}}(t)$, is similar for the shortest and the longest cell lengths (**Appendix 2—figure 9**). In our subsequent analysis, we will therefore assume that either $C(t)$ is constant, or $C(t) \gg K_m$. This implies that $\frac{dL(t)}{dt}$ in **Equation (1)** is directly proportional to $N(t)$.

To derive an expression for $N(t)$, we first note that the old and new cell pole in the cell need to be treated differently. We assume the number of polar TG-sites to saturate within one cellular life-cycle, such that the new pole initiates with $N(t)$ below saturation, while the old pole – inherited from the mother cell – is saturated. Letting the number of TG sites increase proportional to the number of available sites, we arrive at the following kinetic description for $N(t)$

$$\frac{dN(t)}{dt} = \beta(N^{\text{max}} - N(t)) \tag{2}$$

Here, $N^{max}$ is the maximum number of sites at the cell poles, and $\beta$ is a rate constant. This result, together with **Equation (1)**, defines our RAG model. The predicted elongation rates provide a good fit to the experiment for all studied genotypes (**Figure 6E–G**), although the data appear to exhibit a stronger inflection.

Instead of assuming a constant recruitment of TG enzymes, we can construct a more refined model that takes TG recruitment dynamics into account. There is evidence that transglycosylase RodA and PBPs are recruited to the cell pole via the curvature-sensing protein DivIVA (**Letek et al., 2008**; **Sieger et al., 2013**). As shown in **Lenarcic et al., 2009**, DivIVA also recruits itself, leading to the exponential growth of a nucleating DivIVA cluster. Therefore, we let the recruitment rate of TG enzymes be proportional to the number of DivIVA proteins $N_D(t) = N_D(0)e^{\gamma t}$. This results in a modified kinetic description for $N(t)$ (**Equation (2)**):

$$\frac{dN(t)}{dt} = \beta e^{\gamma t}(N^{\text{max}} - N(t)) \tag{3}$$

This refined model can capture more detailed features of the measured elongation rate curves (**Figure 6E–G**), including the stronger inflection, with an additional free parameter, $\gamma$, encoding the self-recruitment rate of DivIVA.

The central assumption of our RAG model is that the growth of the cell poles, mediated via accumulation of TG enzymes, is the rate-limiting step for cellular growth. To test this assumption, we repeated our experiment with a *rodA* knockout (**Sieger et al., 2013**). The SEDS-protein RodA is a mono-functional TG (**Meeske et al., 2016**; **Emami et al., 2017**; **Sjodt et al., 2018**), whose deletion results in a phenotype with a decreased population growth rate in the shaking-flask (**Sieger et al., 2013**). The cells' viability is nonetheless backed up by the presence of bifunctional class A PBPs capable of catalyzing transglycoslyation and transpeptidation reactions. We expect this knockout to lower the efficiency of polar cell wall formation, thus slowing down the rate-limiting step of growth. Specifically, we expect the knockout of *rodA* to mainly affect the efficiency of Lipid-II integration into the murein sacculus. Within our RAG model, this translates to a lowering of the cell wall production per transglycosylase site $\alpha$. This would imply elongation rate curves of similar shape for the $\Delta rodA$ mutant, only scaled down by a factor $\alpha^{WT}/\alpha^{\Delta rodA}$. Indeed, we observe such a scaling down of the elongation rate curves (**Figure 5C**), lending further credence to our model for *C. glutamicum* growth.

A striking feature observed across growth conditions and birth lengths, is the onset of a linear growth regime after approximately 20 min (**Figure 5A–C**). The robustness of this timing can be understood from the RAG model: the regime of linear growth is reached via an exponential decay of the number of available TG sites until saturation is reached. This exponential decay makes the moment of onset of the linear growth regime relatively insensitive to variations in $N(0)$ and $N^{\text{max}}$. Specifically, from **Equation (2)**, it can be shown that the difference between $N(t)$ and $N^{\text{max}}$ is halved every $\ln(2)\beta$ minutes, which amounts to ~8 min given fitted value of $\beta$ (**Appendix 9—table 1**).

Finally, our RAG model makes a prediction for the degree of transglycosylase saturation of the cell poles at birth, relative to the saturation in the linear growth regime. We find that this saturation

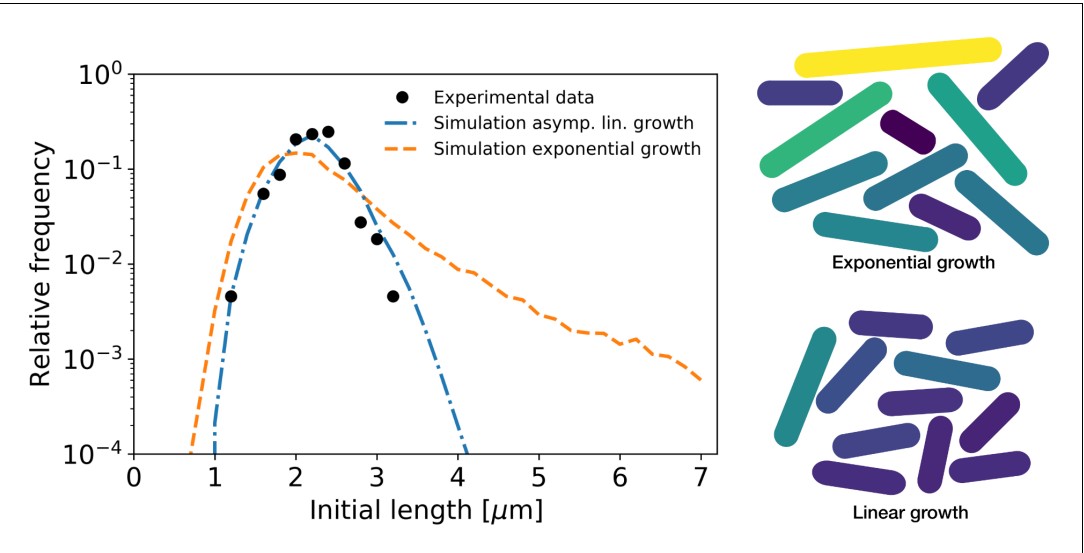

**Figure 7.** Simulation of population growth for asymptotically linear and exponential growth. Left: birth length distribution for simulated asymptotically linear growth (blue dash-dotted line), and for simulated exponential growth (orange dashed line). For both simulations, all relevant growth parameters and distributions are obtained directly from the experimental data. Black dots: experimental birth length distribution. Right: sample of 11 cells from the exponential and asymptotically linear growth simulations, color coded according to length.

is comparable between wild-type and the $\Delta rodA$ mutant (~65% on average), but significantly higher for DivIVA-labeled cells (~80% on average) (*Appendix 9—tables 1* and *2*). Note that the percentage of the saturation levels are relative values and do not suggest that in the DivIVA-mCherry fusion more transglycosylase sites are present in absolute numbers.

## Birth length distribution of linear growers is more robust to single-cell growth variability

After obtaining average single-cell growth trajectories, we next asked how this growth behavior at the single cell level affects the growth of the colony. It was shown that asymmetric division and noise in individual growth times results in a dramatic widening of the cell-size distribution for a purely exponential grower (*Marantan and Amir, 2016*). For an asymptotically linear grower, however, we would expect single-cell variations to have a much weaker impact.

To quantify the difference between asymptotically linear growth and hypothetical exponential growth for *C. glutamicum*, we performed population growth simulations for both cases. For the asymptotically linear growth, we assumed the elongation rate curves obtained from our model. For exponential growth, we assumed the final cell size to be given by $l_d = l_b \exp(\alpha(t_t + \Delta t)) + \Delta l$, with $\alpha$ the exponential elongation rate, $t_t$ the target growth time, $\Delta t$ a time-additive noise term and $\Delta l$ a size-additive noise term. All growth parameters necessary for the simulation were obtained directly from the experimental data (Appendix 10). From this simulation, the distribution of initial cell lengths was determined for each scenario.

The resulting distribution of birth lengths for the asymptotically linear growth case closely matches the experimentally determined distribution (*Figure 7*). By contrast, the distribution for exponential growth is much wider, and exhibits a broad tail for longer cell lengths. This suggests a strong connection between growth mode and the effect of individual growth variations on population statistics. *C. glutamicum* has a high degree of variation of division symmetry (*Appendix 10—figure 1C*) and single-cell growth times, but due to the asymptotically linear growth mode, the population-level variations in cell size are still relatively small. This indicates that linear growth can act as a regulator for cell size.

## Discussion

By developing a novel growth trajectory inference and analysis method, we showed that *C. glutamicum* exhibits asymptotically linear growth, rather than the exponential growth predominantly found in bacteria. The obtained elongation rate curves are shown to be consistent with a model of apical cell wall formation being the rate-limiting step for growth. The RAG model is further validated by experiments with a Δ*rodA* mutant, in which the elongation rate curves look functionally similar, but with a downward shift compared to wild type (*Figure 5B,C*), as expected based on our model. For *C. glutamicum*, apical cell wall formation is a plausible candidate for the rate-limiting step of growth, because synthesis of the highly complex cell wall and lipids for the mycolic acid membrane is cost intensive and a major sink for energy and carbon in *Corynebacteria* and *Mycobacteria* (*Brennan, 2003*).

An analysis of elongation rates as a function of time and birth length has previously been done in *B. subtilis* by binning cells based on birth length (*Nordholt et al., 2020*). Applying this method to our data set yields elongation rates averaged over cells within a binning interval (*Appendix 2—figure 5*). Averaging our inferred elongation rates over the same bins, we find the two methods to yield consistent results. The binning method, however, involves a tradeoff: a smaller bin width results in a larger error on the inferred elongation rates, whereas a larger bin width averages out all variation within a larger birth length interval. Our method does not suffer from this binning-related tradeoff, and it provides detailed elongation rate curves at any given birth length. In other recent work (*Kar et al., 2021*), average growth rate curves were calculated as a function of cell phase. Our method provides additional detail by extracting the dependence of elongation rate on birth length as well as time since birth.

Our proposed growth model shares some similar features to recent experimental observations on polar growth in Mycobacteria (*Hannebelle et al., 2020*). Polar growth was shown to follow 'new end take off' (NETO) dynamics (*Hannebelle et al., 2020*), in which the new cell pole makes a sudden transition from slow to fast growth, leading to a bilinear polar growth mode. In our proposed growth model for *C. glutamicum* however, the new pole gradually increases its average elongation rate before saturating to a constant maximum. The deviation of *C. glutamicum* from NETO dynamics can also be seen by comparing each of the pole intensities in the HADA staining experiment, which does not show any signatures of NETO-like growth (*Appendix 2—figure 11*). It remains unclear which molecular mechanisms produce the differences in growth between such closely related species. However, the mode of growth described here for *C. glutamicum* might well be an adaption to enable higher growth rates.

To investigate the implications of our inferred single-cell growth mode for cell-size homeostasis throughout a population of cells, we performed simulations of cellular growth and division over many generations. We found that our asymptotically linear growth model accurately reproduces the experimental distribution of cell birth lengths. By contrast, a model of exponential growth predicts a much broader distribution with a long tail for larger birth lengths. This indicates a possible connection between mode of growth and permissible growth-related noise levels for the cell. Indeed, if single-cell growth variability is reduced by a factor 3, the distributions corresponding to both growth modes show a similarly narrow width (*Appendix 10—figure 2*). However, an asymptotically linear grower is able to maintain a narrow distribution of cell sizes even for higher noise levels, whereas for an exponential grower this distribution widens dramatically (*Figure 7*).

The enhanced robustness of the length distribution of linear growers is interesting from an evolutionary point of view. Most rod-shaped bacteria use sophisticated systems, such as the Min system, to ensure cytokinesis precisely at midcell (*Bramkamp et al., 2009*; *Lutkenhaus, 2007*). Bacteria encoding a Min system grow by lateral cell wall insertion. In contrast, rod-shaped bacteria in the *Actinobacteria* phylum such as *Mycobacterium* or *Corynebacterium* species, grow apically and do not contain a Min system, nor any other known division site selection system (*Donovan and Bramkamp, 2014*). *C. glutamicum* rather couples division site selection to nucleoid positioning after chromosome segregation via the ParAB partitioning system (*Donovan et al., 2013*), and has a broader distribution of division symmetries. We speculate that due to *C. glutamicum*'s distinct growth mechanism, a more precise division site selection mechanism is not necessary to maintain a narrow cell size distribution.

The elongation rates reported in this work reflect the increase in cellular volume over time. However, the increase in cell *mass* is not necessarily proportional to cellular volume. In exponentially growing *E. coli*, the cellular density was recently reported to systematically vary during the cell cycle, while the surface-to-mass ratio was reported to remain constant (*Oldewurtel et al., 2019*). It is unknown how single-cell mass increases in *C. glutamicum*, but it would follow exponential growth if mass production is proportional to protein content. This raises the question how linear volume growth and exponential mass growth are coordinated. The presence of a regulatory mechanism for cell mass production that couples to cell volume is implied by the elongation rate curves obtained for the Δ*rodA* mutant. As the elongation rate is lower in this mutant, average mass production needs to be lowered compared to the WT in order to prevent the cellular density from increasing indefinitely.

Our growth trajectory inference method is not cell-type specific, and can be used to obtain detailed growth dynamics in a wide range of organisms. The inferred asymptotically linear growth of *C. glutamicum* deviates from the predominantly found exponential single-cell bacterial growth, and suggests the presence of novel growth regulatory mechanisms.

# Materials and methods

## Key resources table

| Reagent type (species) or resource | Designation | Source or reference | Identifiers | Additional information |
|---|---|---|---|---|
| Gene (include species here) | 'divIVA'; 'rodA' | KEGG | 'cg2361'; 'cg0061' | |
| Strain, strain background (*Corynebacterium glutamicum*) | 'ATCC 13032'; 'RES 167' | 'ATCC'; '*Tauch et al., 2002*' | '13032';"RES 167' | |
| Genetic reagent (*Corynebacterium glutamicum*) | 'RES 167 divIVA::divIVA-mCherry';"RES 167 Δ rodA, divIVA::divIVA-mCherry' | '*Donovan et al., 2012*'; '*Sieger et al., 2013*' | 'CDC010'; 'BSC002' | |
| Chemical compound, drug | HADA stain | Tocris Bioscience | 6647/5 | |
| Software, algorithm | MorpholyzerGT | This paper | | see Materials and methods |
| Other | CellASIC microfluidic System | Millipore | B04A | |

## Culture and live-cell time-lapse imaging

Exponentially growing cells of *C. glutamicum WT*, *C. glutamicum divIVA::divIVA-mCherry* and *C. glutamicum divIVA::divIVA-mCherry ΔrodA* respectively, grown in BHI–medium (Oxoid) at 30°C and 200 rpm shaking, were diluted to an $OD_{600}$ of 0.01. According to the manufacturer's manual cells were loaded into a CellASIC- microfluidic plate type B04A (Merck Milipore) and mounted on a Delta Vision Elite microscope (GE Healthcare, Applied Precision) with a standard four-color InSightSSI module and an environmental chamber heated to 30°C. Images were taken in a three-minute interval for 10 hr with a 100×/1.4 oil PSF U-Plan S-Apo objective and a DS-red-specific filter set (32% transmission, 0.025 s exposure).

## Staining of newly inserted peptidoglycan and visualization in demographs

For the staining of nascent PG, 1 ml of exponentially growing *C. glutamicum ATCC 13032* cells, cultivated in BHI–medium (Oxoid) at 30°C and 200 rpm, were harvested, washed with PBS and resuspended in 25 µl PBS, together with 0.25 µl of 5 mM HADA dissolved in DMSO. The cells were incubated at 30°C in the dark for 5 min, followed by a two-time washing step with 1 ml PBS and finally resuspended in 100 µl PBS. To obtain still- phase-contrast and fluorescent micrographs, 2 µl of the cell suspension were immobilized on an agarose pad. For microscopy, an Axio Imager (Zeiss) equipped with EC Plan-Neofluar 100x/1.3 Oil Ph3 objective and a Axiocam camera (Zeiss) was used together with the appropriate filter sets (ex: 405 nm; em: 450 nm). For single-cell analysis and the

visualization in demographs, custom algorithms, developed in FIJI and R (*Schindelin et al., 2012*; *R Development Core Team, 2003*), were used. The code is available upon request.

### Image analysis

For image analysis, a custom-made algorithm was developed using the open-source programs FIJI and R (*Schindelin et al., 2012*; *R Development Core Team, 2003*). During the workflow unique identifiers to single-cell cycles are assigned. The cell outlines are determined manually. Individual cells per timeframe are extracted then from the raw image and further processed automatically. The parameters length, area and relative septum position are extracted and stored together with the genealogic information and the timepoint within the respective cell cycle. The combination of image analysis and cell cycle dependent data structuring yields a list that serves as a base for further analysis. The documented code is available at: https://github.com/Morpholyzer/MorpholyzerGeneration-Tracker (copy archived at swh:1:rev: d01d362ea53b9be6027f29fb85668a0ed418398a, *Morpholyzer, 2021*).

## Acknowledgements

This work was further funded by grants from the Deutsche Forschungsgemeinschaft (project P05in TRR174, granted to MB and project P06 in TRR174, granted to CB). JM is supported by a DFG fellowship within the Graduate School of Quantitative Biosciences Munich (QBM). We thank our colleagues from CB and MB groups for discussions, feedback and comments on the manuscripts.

## Additional information

### Funding

| Funder | Grant reference number | Author |
|---|---|---|
| Ludwig-Maximilians-Universität München | Graduate Student Stipend | Joris JB Messelink |
| Deutsche Forschungsgemeinschaft | TRR 174 project P06 | Joris JB Messelink Chase P Broedersz |
| Deutsche Forschungsgemeinschaft | TRR 174 project P05 | Fabian Meyer Marc Bramkamp |

The funders had no role in study design, data collection and interpretation, or the decision to submit the work for publication.

### Author contributions

Joris JB Messelink, Software, Formal analysis, Investigation, Visualization, Writing - original draft, Writing - review and editing; Fabian Meyer, Data curation, Investigation, Visualization, Methodology, Writing - original draft, Writing - review and editing; Marc Bramkamp, Chase P Broedersz, Conceptualization, Supervision, Writing - original draft, Writing - review and editing

### Author ORCIDs

Joris JB Messelink ![ORCID] https://orcid.org/0000-0002-7986-4527
Fabian Meyer ![ORCID] https://orcid.org/0000-0002-8305-0390
Marc Bramkamp ![ORCID] https://orcid.org/0000-0002-7704-3266
Chase P Broedersz ![ORCID] https://orcid.org/0000-0001-7283-3704

### Decision letter and Author response

Decision letter https://doi.org/10.7554/eLife.70106.sa1
Author response https://doi.org/10.7554/eLife.70106.sa2

## Additional files

### Supplementary files
- Source data 1. HADA staining data.
- Source data 2. Elongation measurement data.
- Transparent reporting form

### Data availability
All data generated during this study are included in the manuscript and supporting files.

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

# Appendix 1

## Single-cell growth mode for apical cell wall formation as a rate-limiting step for growth

To study growth limited by polar cell wall formation, we start by considering the Michaelis-Menten equation describing this formation process (Main Text *Equation (1)*):

$$\frac{dL(t)}{dt} = \alpha \frac{C(t)N(t)}{K_m + C(t)}, \tag{A1}$$

with $L(t)$ the cell length at time $t$, $C(t)$ the concentration of cell wall building blocks in the cytosol, $N(t)$ the number of transglycosylases at the cell pole, $K_m$ the Michaelis constant for this reaction, and $\alpha$ a proportionality constant.

In Main Text *Figure 1*, we consider two scenarios. (1) Abundant availability of cell wall building blocks, that is $C(t) \gg K_m$, and (2) scarcity of cell wall building blocks, that is, $C(t) < K_m$.

### A1.1 Building block insertion as a rate-limiting step for growth

In scenario (1), *Equation (A1)* reduces to $\frac{dL(t)}{dt} = \alpha N(t)$. In the regime of a constant number of transglycosylases at the pole, this implies that $\frac{dL(t)}{dt}$ is constant, resulting in linear growth.

### A1.2 Building block availability as a rate-limiting step for growth

In scenario (2), the dynamics of building block creation, usage, and dilution need to be considered to determine the cellular elongation rate behavior. For the number of building blocks in the cytosol as a function of time $n(t)$, we can write the following differential equation:

$$\frac{dn(t)}{dt} = aV(t) - b\frac{dV(t)}{dt}. \tag{A2}$$

Here, $a$ encodes building block production rate per unit volume, and $b$ encodes building block usage by the cell wall formation mechanism, making use of $\frac{dA(t)}{dt} \propto \frac{dV(t)}{dt}$. To connect *Equation (A2)* to *Equation (A1)*, we note that $C(t) = \frac{n(t)}{V(t)}$. Restricting ourselves to the regime $C(t) \ll K_m$, we can rewrite *Equation (A1)* to

$$\frac{dV(t)}{dt} = c\frac{n(t)}{V(t)}, \tag{A3}$$

where we made use of $\frac{dL(t)}{dt} \propto \frac{dV(t)}{dt}$. Here, $c$ encodes the proportionality between volume increase and the concentration of building blocks.

Combining *Equation (A2)* with *Equation (A3)*, we obtain a set of coupled nonlinear differential equations governing the time-evolution of $V(t)$. These equations have no simple analytic solution; however, we can numerically explore the dependence of $V(t)$ on the differential equation parameters. To do this, we first absorb $c$ into $n(t)$, leaving us with two free parameters and two boundary conditions. The boundary conditions we set by imposing $V(0) = 1$ and $V(1) = 2$. In *Appendix 1—figure 1A*, we see that depending on the choice for $a$ and $b$ we can have either sublinear, approximately linear, or superlinear growth. This demonstrates that the single-cell growth mode is dependent on the physiology of building block creation and depletion in the cell.

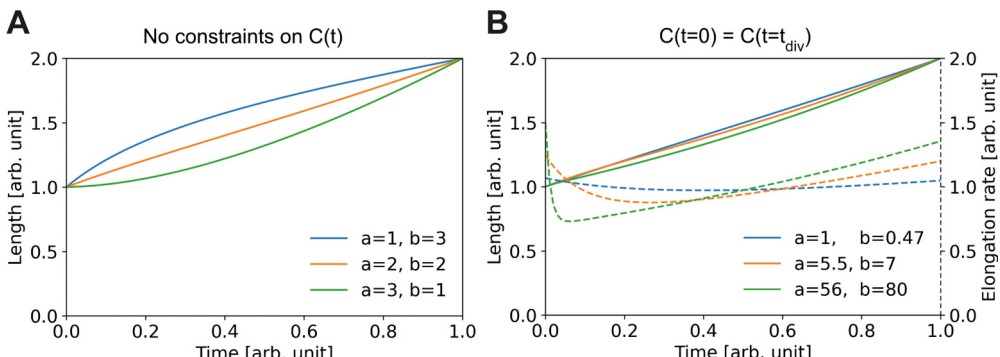

**Appendix 1—figure 1.** Elongation curves assuming building block availability is the limiting step for growth. (**A**) Numerically obtained solutions for $V(t)$, from the set of coupled differential equations *Equation (A2)* and *Equation (A3)*. For all solutions, $V(0) = 1$ and $V(1) = 2$ are imposed. (**B**) Solutions as in (**A**), but with the additional constraint that the concentrations before and after division are the same, i.e. $C(t = 0) = C(t = t_{\mathrm{div}})$. Solid lines: solutions for $V(t)$. Dashed lines: corresponding $\frac{dV(t)}{dt}$, which are proportional to the concentration $C(t)$ per *Equation (A3)*.

We can further constrain the solution space by demanding that the concentration of building blocks $C(t) = \frac{n(t)}{V(t)}$ is the same at birth and division. In this scenario, the observed variation in elongation curves is smaller (solid lines *Appendix 1—figure 1B*), however the corresponding elongation rates (dashed lines *Appendix 1—figure 1B*) still show marked qualitative differences between parameter choices.

## Appendix 2

### Supplementary figures

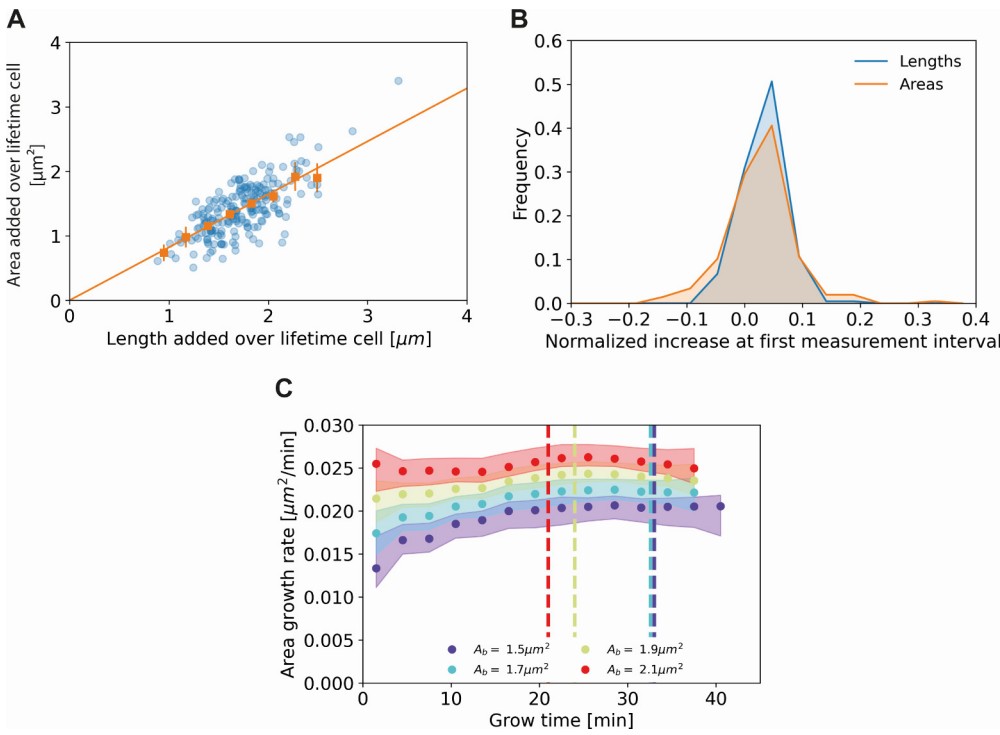

**Appendix 2—figure 1.** Comparing cell length and cell area measurements. (**A**) Length added versus area added over the cell lifetime for all cells included in our analysis (blue dots), together with averaged values at 0.2 µm intervals (orange squares) and 95% confidence intervals (orange vertical lines). The results are consistent with a proportional relationship (orange line). (**B**) Histogram of the normalized increase at first measurement interval using cell lengths (blue) and areas (orange). For the cellular lengths, this quantity is defined as $\frac{L(t=3min)-L(t=0)}{\langle l_b \rangle}$, whereas for the areas it is defined as $\frac{A(t=3min)-A(t=0)}{\langle A_b \rangle}$, with $A(t)$ the area at time $t$ and $A_b$ the birth area. The wider distribution for the areas suggests a higher measurement noise for this quantity. (**C**) Area growth rate for DivIVA-labeled cells using estimated cell areas. The trajectories are consistent with those obtained from cell lengths (Main Text *Figure 5A*).

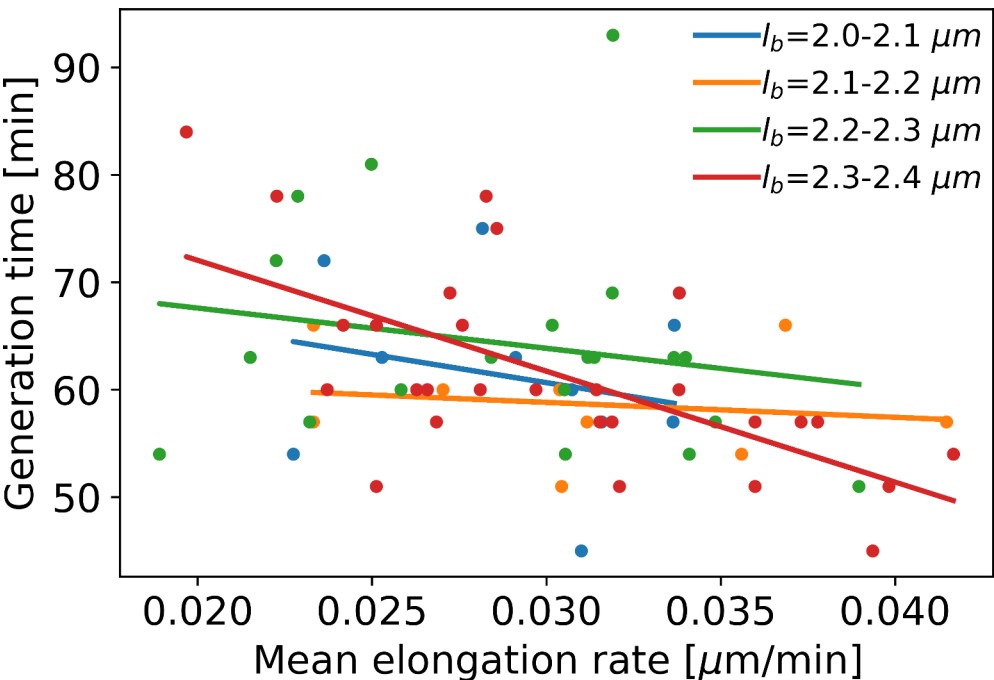

**Appendix 2—figure 2.** Mean elongation rate versus generation time for cells in four different birth size bins. Linear fits are indicated by solid lines. As generation times within a birth size bin tend to be shorter for faster-growing cells, the elongation rate curves obtained with our method become biased after the first division event. This justifies only using the part of the elongation rate curves until the first division event for further analysis.

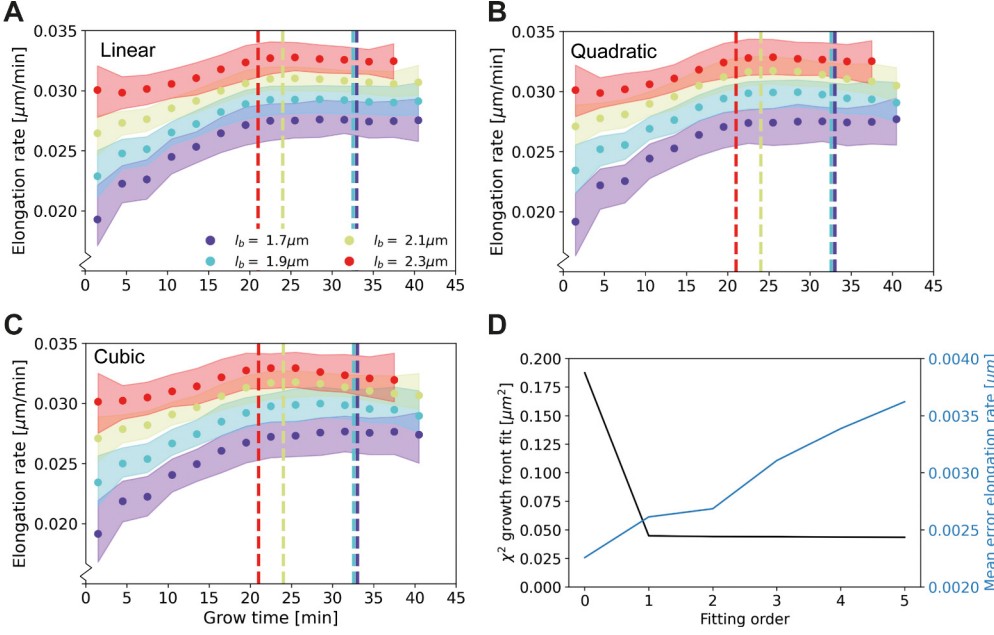

**Appendix 2—figure 3.** Elongation rate curves for different orders of the wave front fit of Main Text *Figure 3A*: Linear (**A**), quadratic (**B**), and cubic (**C**). (**D**) $\chi^2$ of the fit of the wave front of Main Text *Figure 3A* for different fitting orders, together with the mean error on the elongation rate curves.
*Appendix 2—figure 3 continued on next page*

The negligible improvement of the goodness-of-fit after the first order justifies the use of a linear fit for further analysis.

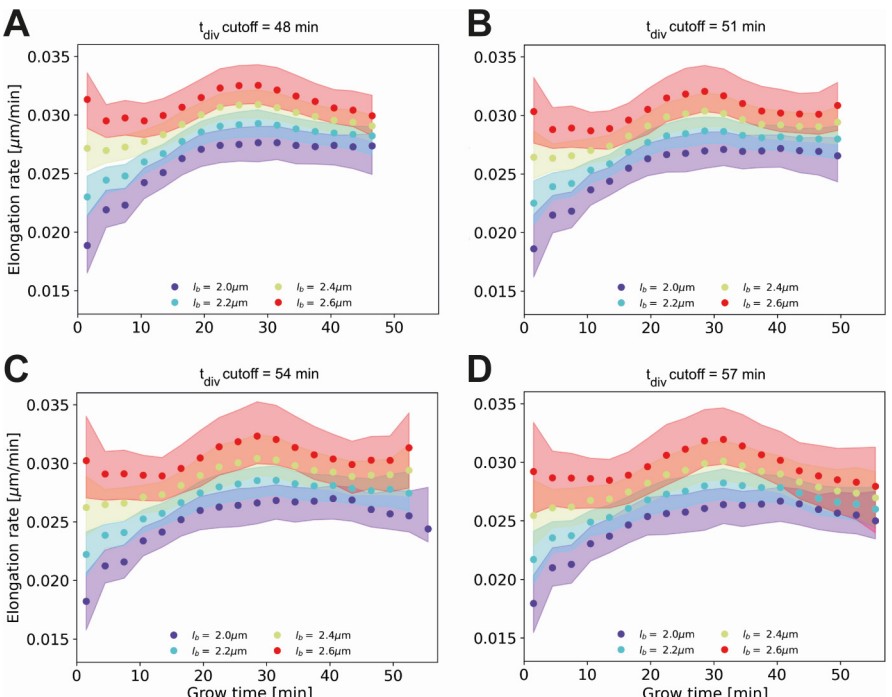

**Appendix 2—figure 4.** Conditional elongation rate curves, conditioned on DivVIA-labeled cells that have a generation time larger than a set cutoff value: 48 min (**A**), 51 min (**B**), 54 min (**C**) and 57 min (**D**). The inferred elongation rate curves still display similar growth behavior to the unconditioned population (Main Text *Figure 5A*), but exhibit an overall downwards shift with increasing cutoff times. For larger cutoff times, the number of cells included decreases, resulting in larger errors on the inferred elongation rates. The linear growth phase observed until the cutoff time for the unconditioned population is seen to persist for longer grow times.

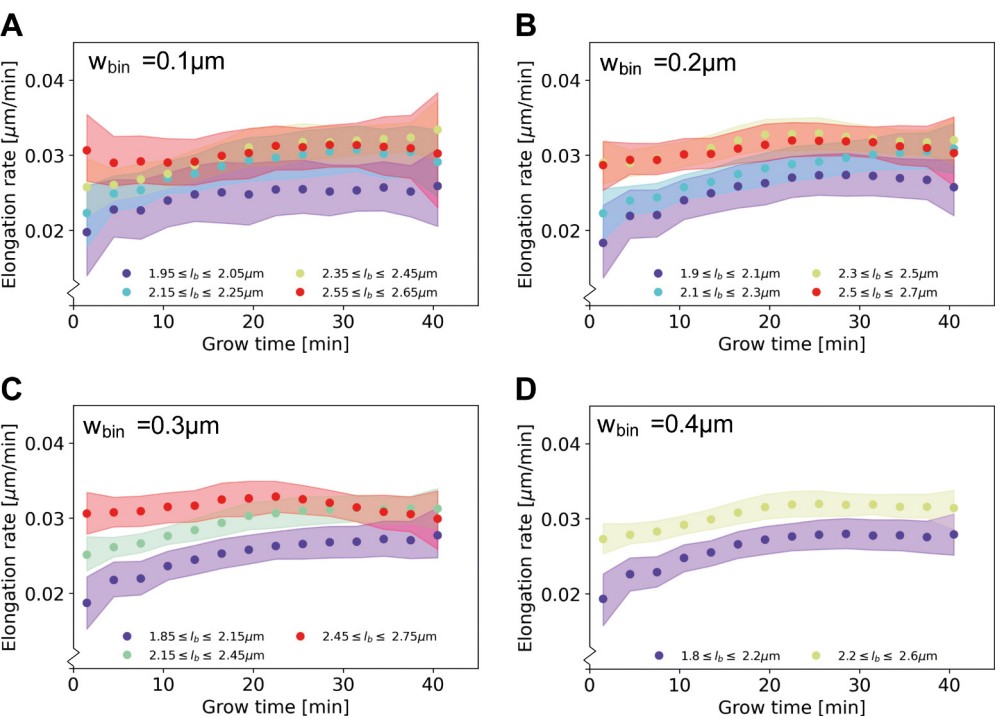

**Appendix 2—figure 5.** Elongation rate curves obtained through a binning procedure. Cells are divided into birth length bins, and for each bin the average length as a function of grow time is calculated. The resulting elongation curves are smoothened according to the same procedure as the elongation curves presented in the main text (see Appendix 5). From the smoothened elongation curves, elongation rates are calculated as a function of grow time. Results are shown for a bin width of 0.1 μm (**A**), 0.2 μm (**B**), 0.3 μm (**C**), 0.4 μm (**D**), where each $l_b$ indicates the center of the birth length bin.

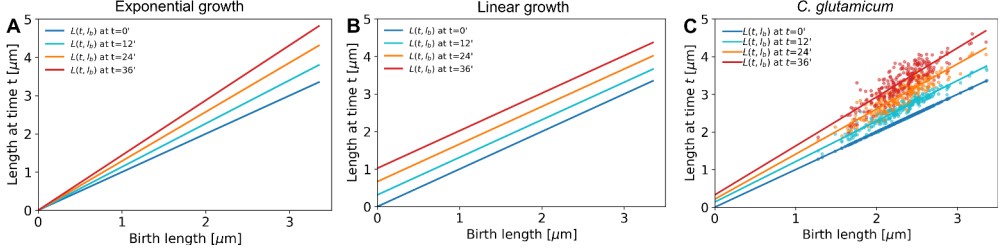

**Appendix 2—figure 6.** A linear fit through the cell lengths at each time step would be enough to describe exponential growth (**A,** offset is zero for all time stamps) as well as linear growth (**B,** slope is equal to 1 for all time stamps). *C. glutamicum* (**C**) matches neither of these growth modes.

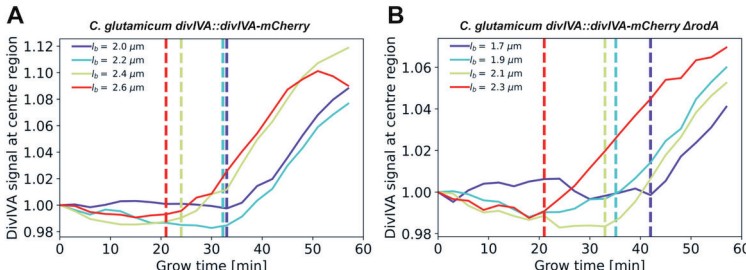

**Appendix 2—figure 7.** The average DivIVA-mCherry signal from the cell center over time is shown for DivIVA-labeled cells (**A**) and Δ*rodA* DivIVA-labeled cells (**B**). The cell center is here defined as the region between 20% and 80% of the total cell length. The onsets of septum formation, derived from the DivIVA signal-mCherry signal, are indicated by the dashed lines; these do not consistently coincide with the levelling off of elongation rates (Main Text *Figure 5A*). This is inconsistent with the leveling off being due to a competition between polar growth and septum formation.

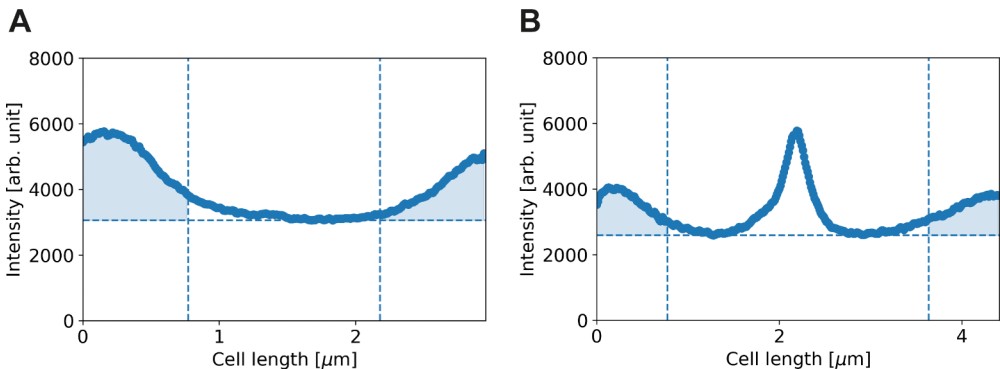

**Appendix 2—figure 8.** Calculation of corrected polar HADA intensity, illustrated for two HADA pro-files. Solid line: HADA intensity profile. Dashed horizontal line: minimum of HADA profile. Dashed vertical lines: boundary of polar region. Shaded area: calculated total polar intensity. Results shown for a cell with a length of 2.3 μm (**A**) and 4.4 μm (**B**).

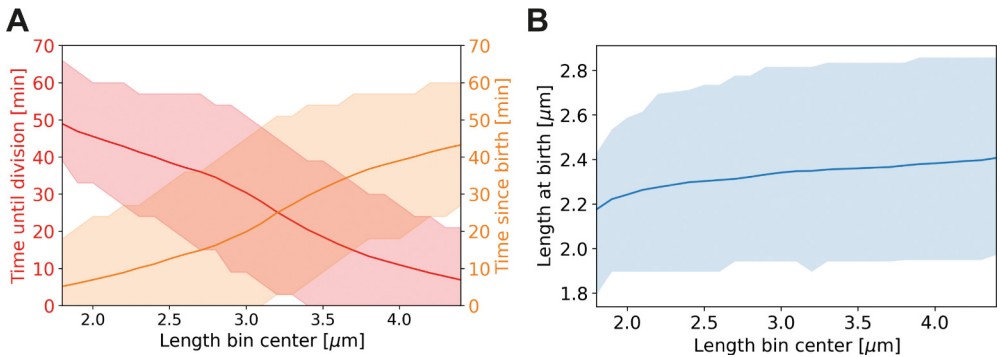

**Appendix 2—figure 9.** Average properties of wild-type cells as a function of length. Values are shown over the range of observed lengths in the HADA staining experiment, using a moving average with the same width (±0.7 μm) as in Main Text *Figure 6C*. (**A**) Red line: average time until division, together with the two standard deviation bounds (red shaded area). Orange line: average

*Appendix 2—figure 9 continued on next page*

*Appendix 2—figure 9 continued*

time since birth, together with two standard deviation bounds (orange shaded area). (**B**) Blue line: average birth length for each birth length bin (blue line), together with the two standard deviation bounds (blue shaded area).

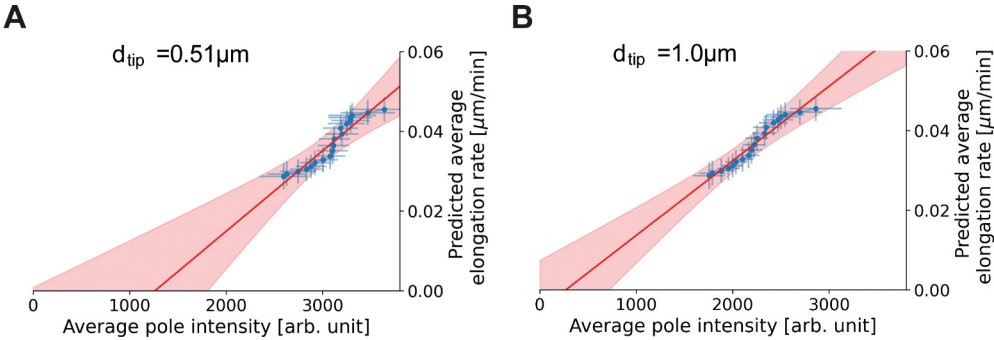

**Appendix 2—figure 10.** Proportionality between average pole intensity and predicted average elongation rate for different polar region definitions. Average elongation rate as a function of cell length (red), predicted from obtained average elongation rate curves, together with the average HADA staining intensity at the cell pole after background correction (blue). Results are shown for a polar region defined to be within 0.51 μm (**A**) and 1.0 μm (**B**) of the cell tip.

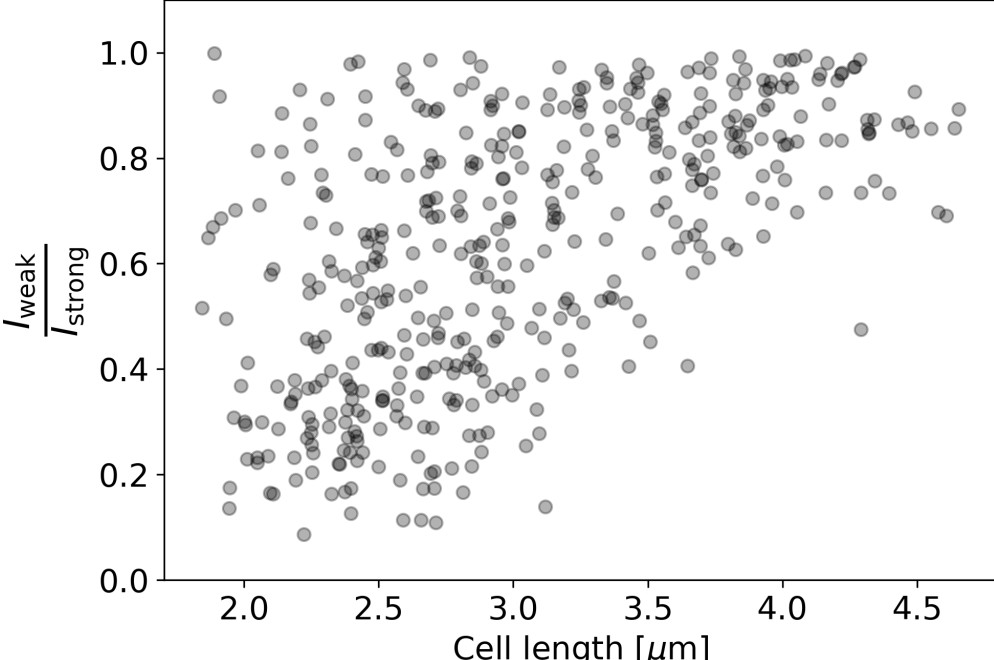

**Appendix 2—figure 11.** Ratio of intensities between the weaker and the stronger pole of each cell in the HADA staining experiment. Polar intensities are calculated as described in *Appendix 2—figure 8*. Here, $I_{weak}$ denotes the intensity of the cell pole with the weaker HADA intensity signal, and $I_{strong}$ denotes the intensity of the pole with the stronger signal. For NETO-like growth (*Hannebelle et al., 2020*), a clustering of values around 0 (before new end take off) and 1 (after new end take off) would be expected, which is not observed here.

## Appendix 3

### Measurement noise estimate

To obtain an estimate for the measurement noise from our time-series growth data, we make use of length measurements at subsequent time intervals. For short enough time intervals, the variance of the length differences between intervals can be used as a measure of the measurement noise. However, since we expect cellular growth to also significantly contribute to this variance within the 3 min measurement interval, we have to separate out the two contributions.

To separate out the two contributions to the variance in subsequent length measurements, we write this variance as

$$\text{Var}(l_m(t+\Delta t) - l_m(t)) \;=\; \text{Var}(l(t+\Delta t) - l(t)) + 2\sigma_n^2 \tag{A7}$$

with $l_m(t)$ the measured length at time $t$, $l(t)$ the actual length at time $t$, and $\sigma_n$ the standard deviation of the measurement noise. This expression can be derived by noting that for a single elongation trajectory, we have

$$l_m(t+\Delta t) - l_m(t) \;=\; l(t+\Delta t) + \xi - (l(t) + \xi) = l(t+\Delta t) - l(t) + \sqrt{2}\xi, \tag{A8}$$

with $\xi$ the measurement noise. A solution for $\sigma_n$ can be found if the functional form of $\text{Var}(l(t), l(t+\Delta t))$ is known, by obtaining values for multiple $\Delta t$ and treating $\sigma_n$ as a fitting parameter. To obtain this functional form, we make use of the observed linear growth regime after ~20 min (Main Text *Figure 5*). We observe that the elongation rate is approximately constant in this regime for cells of all birth lengths, and now assume that this is also true for cells individually within this regime. The contrary would imply that non-constant single-cell elongation rates precisely cancel out across time and birth lengths to produce linear growth, which seems biologically implausible.

For linearly growing single cells, the standard deviation of $l(t+\Delta t) - l(t)$ is proportional to $\Delta t$, implying that the term $\text{Var}(l(t), l(t+\Delta t))$ is of the form

$$\text{Var}(l(t+\Delta t) - l(t)) = c\Delta t^2, \tag{A9}$$

with $c$ an unknown parameter. To simultaneously obtain $c$ and $\sigma_n$, we fit *Equation (A7)* under substitution of *Equation (A9)* to the DivIVA-labeled cell data over the regime between the onset of linear growth (18 min, black dashed line *Appendix 3—figure 1*) and the first division event (36 min, gray dashed line *Appendix 3—figure 1*). From this fit, we obtain the estimates $\sigma_n = 0.060 \pm 0.018\ \mu\text{m}$ and $c = 4.5x10^{-5} \pm 0.47\text{xm}^2\ \text{min}^{-2}$, where the error margins are determined via bootstrapping. This value of $\sigma_n$ is used in the correction procedure for assigned birth lengths described in Appendix 4.

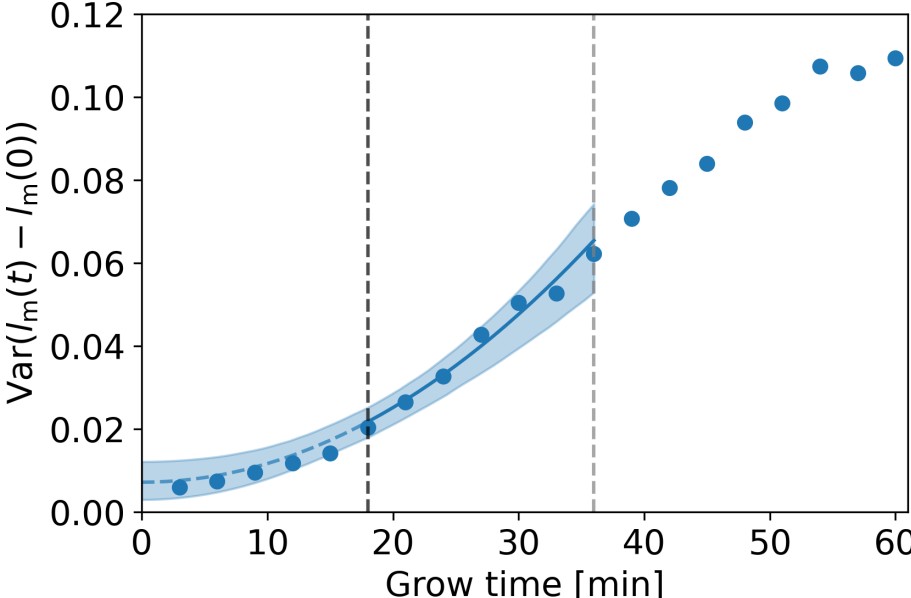

**Appendix 3—figure 1.** Estimation of measurement noise procedure. Blue dots: variance of $l_m(t) - l_m(0)$ as a function of grow time for the DivIVA-labeled cells, with $l_m(t)$ the measured cellular length at grow time $t$. A fit of *Equation (A7)* under substitution of *Equation (A9)* (blue line) is made to the points between the onset of linear growth (black dashed line) and the moment of first division (gray dashed line). The value of the extrapolated fit (blue dashed line) at t=0 is equal to $2\sigma_n^2$, with $\sigma_n$ the standard deviation of the measurement noise. The 95% confidence intervals of the model fit (blue shaded area) are obtained via bootstrapping.

## Appendix 4

### Bias correction procedure for assigned birth lengths

Before calculating average elongation rate curves, a statistical bias arising in the assignment of birth lengths to each curve needs to be corrected for. This bias is not specific to the inference method introduced in this paper, but arises in any procedure involving the assignment of lengths to a cells within a population, if there is noise in the measurement of individual cell lengths.

Due to measurement noise, cells will be assigned to birth lengths that systematically differ from their actual birth lengths. Specifically, given that the birth lengths in the population follow a symmetric, unimodal distribution, cells with a measured birth length larger than the population mean will on average be assigned a birth length that is larger than their actual length. Conversely, cells with a birth length smaller than the population mean will on average be assigned a birth length that is smaller.

The magnitude of the systematic deviation in the assignment of birth lengths is calculated as follows. Given that the cellular birth lengths follow a Gaussian distribution $P_l(l_b)$ with mean $\mu_l$ and standard deviation $\sigma_l$, and the measurement noise follows a Gaussian distribution $P_n(\Delta l)$ with mean 0 and standard deviation $\sigma_n$, the distribution of measured lengths will again be a Gaussian, with mean $\mu_m = \mu_l$ and standard deviation $\sigma_m = \sqrt{\sigma_l^2 + \sigma_n^2}$.

For a given measured birth length $l_m$, we now consider the probability distribution of corresponding actual birth lengths $P_l(l_b|l_m)$. This distribution is given by

$$P_l(l_b|l_m) = P_l(l_b)P_n(l_m - l_b). \tag{A4}$$

The product of two Gaussian distributions is again Gaussian, with a mean equal to

$$\langle l_b|l_m \rangle = \frac{\sigma_n^2 \mu_l + \sigma_l^2 \int l_b P_n(l_m - l_b)\mathrm{d}l_b}{\sigma_n^2 + \sigma_l^2} = \frac{\sigma_n^2 \mu_l + \sigma_l^2 l_m}{\sigma_n^2 + \sigma_l^2}. \tag{A5}$$

*Equation (A5)* thus provides the transformation needed to remove the systematic bias in the assignment of birth lengths, and to determine the most likely birth length $l_b$ to a cell with a measured birth length $l_m$. For an estimation of the experimental measurement noise, see Appendix 3.

For the length increase since birth, there is no systematic bias once the bias in birth length has been removed. We can see this as follows. For each single-cell elongation trajectory, the measured length $l_m(t)$ at time $t$ is given by

$$l_m(t) = l_b + \Delta l_t + \xi, \tag{A6}$$

with $\xi$ the measurement noise and $\Delta l_t$ the length increase since birth at time $t$. As the measurement noise $\xi$ has a zero mean, there is no systematic bias in length increases after birth, provided that we have an unbiased estimate for the birth length $l_b$.

To test the derived correction procedure for assigned birth lengths, we performed a simulation of a population of growing cells, with the length measurement subject to noise. The measurement noise was sampled from a Gaussian, with the same standard deviation as estimated for experiment (Appendix 3). The single-cell growth mode was chosen as an input parameter. We analyzed two choices for input growth mode: linear (*Appendix 4—figure 1A,C*, dashed lines) and exponential (*Appendix 4—figure 1B,D*, dashed lines), with elongation rates comparable in magnitude to measured elongation rates.

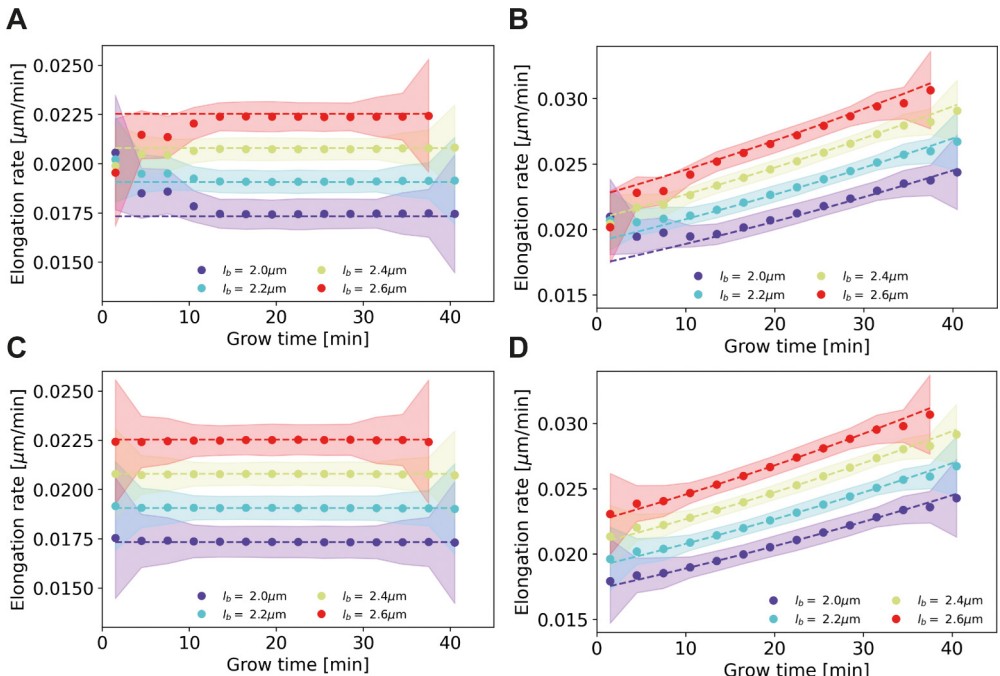

**Appendix 4—figure 1.** Elongation rate inference on simulated data sets, with and without bias correction procedure for assigned birth lengths. For all panels: dashed lines: input elongation rates. Dots: mean inferred average elongation rates, obtained by applying our inference procedure to 1000 simulated data sets. Shaded areas: 2σ bounds on the inferred elongation rates. For all simulated data sets, the measurement noise is drawn from a Gaussian distribution with a standard deviation of 0.075 μm, matching the estimated experimental noise (Appendix 4). The population size and birth length distribution are chosen to match those observed for the DivIVA-labeled cells. Simulation conditions: (**A**) Linear input elongation rates constructed by setting $l(t) = l_b + 0.26 l_b t$. No bias correction procedure for assigned birth lengths is applied. (**B**) Exponential input elongation rates constructed by setting $l(t) = l_b e^{0.26t}$. No bias correction procedure for assigned birth lengths is applied. (**C**) Input elongation rates as in (**A**). The bias correction procedure for assigned birth lengths is applied. (**D**) Input elongation rates as in (**B**). The bias correction procedure for assigned birth lengths is applied.

For each single-cell growth mode, we applied our elongation rate inference procedure to simulated cell lengths subject to measurement noise. Without correcting for a bias in assigned birth lengths, we find a systematic deviation between inferred elongation rates and input elongation rates in both cases (*Appendix 4—figure 1A,B*). With the implementation of the correction for assigned birth lengths, the input elongation rates are, however, accurately recovered (*Appendix 4—figure 1C,D*).

Minor deviations from the input elongation rates can still be seen for exponentially growing cells (*Appendix 4—figure 1D*), arising from applying a Gaussian smoothing to elongation curves that are locally nonlinear due to limited time resolution. However, this effect is small compared to the uncertainty on the inferred elongation rates.

## Appendix 5

### Smoothing of elongation curves

We obtain elongation rate curves (Main Text *Figure 5* and *Figure 6C*) by taking a numerical derivative of smoothed growth trajectories. For the smoothing, a Gaussian smoothing procedure was used. In this procedure, a moving average is applied twice over groups of three subsequent time stamps of average elongation curves. As a check of the validity of the smoothing procedure, we also compare elongation rates before and after smoothing (*Appendix 5—figure 1*).

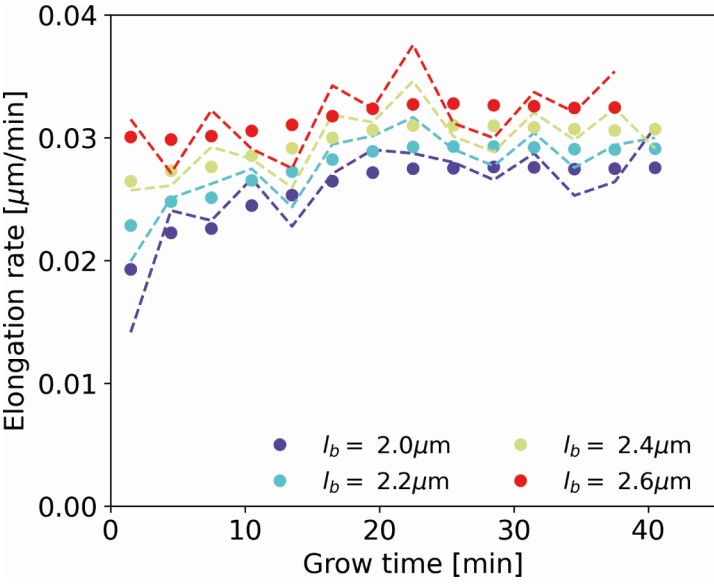

**Appendix 5—figure 1.** Average elongation rate curves obtained after Gaussian smoothing of the inferred average elongation curves (dots), together with average elongation rate curves obtained from unsmoothed average elongation curves (dashed lines).

## Appendix 6

### Calculating mean elongation curves as a function of time until division

The construction of the average elongation curves $L(t - t_d, l_d )$ as a function of the time until division $t - t_d$ and division length $l_d$ is as follows. We relate the length at time $t - t_d$ to the division length $l_d$ for all cells, and use linear fits to obtain a family of curves $L_{t-t_d}(l_d)$ for each $t - t_d$. From this family of relations $L_{t-t_d}(l_d)$, we can subsequently compute $L(t - t_d, l_d )$ for any choice of $l_d$. The resulting mean elongation rate curves are shown in *Appendix 6—figure 1*.

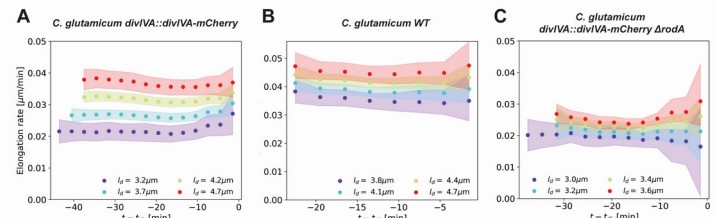

**Appendix 6—figure 1.** Inferred elongation rates as a function of the time until division, shown for DivIVA-labeled cells (**A**), wild-type cells (**B**) and the Δ*rodA* mutant (**C**). To obtain these curves, the elongation rate inference procedure described in the Main Text was applied, with the modification that $L(t - t_{\mathrm{div}}, l_d )$ was calculated, rather than $L(t, l_b )$. This yields average elongation rate curves as a function of division length, which are unbiased until the growth time of the shortest-lived cell (left endpoints of the elongation rate curves). The inferred linear growth regime for later grow times persists until division.

## Appendix 7

### Testing the elongation rate inference procedure

To test our elongation rate inference procedure, we generated a simulated data set with elongation rates as inferred by our inference procedure for DivIVA-labeled cells (Main Text *Figure 5*). The distribution of birth lengths and division lengths of the simulated cells are taken to match the experimentally observed distributions. On each simulated data point, a measurement noise as determined in Appendix 3 is applied. On the simulated data set subject to noise, we apply the assigned birth length correction procedure as described in Appendix 4, and subsequently apply our elongation rate inference procedure. We find that the input elongation rates are accurately recovered (*Appendix 7—figure 1*), demonstrating the internal consistency of our inference approach.

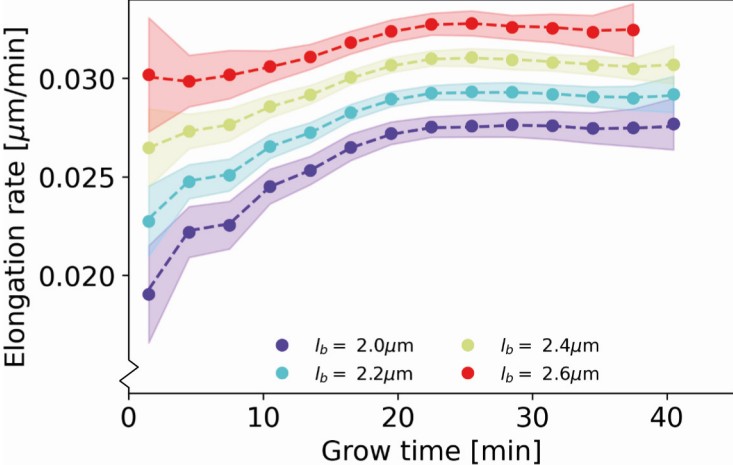

**Appendix 7—figure 1.** Recovery of inferred elongation rates from simulated growth Dashed lines: input elongation rates, as inferred for DivIVA-labeled cells (Main Text *Figure 5A*). Dots: average of elongation rates inferred from simulated growth experiment. Shaded areas: 95% confidence intervals inferred from simulated growth experiment, obtained via bootstrapping.

## Appendix 8

### Prediction of average elongation rate as a function of cell length

To calculate the predicted average elongation rates shown in Main Text *Figure 6C*, we make use of our time-series data for wild-type cells, and the inferred mean elongation rates shown in Main Text *Figure 5B*.

We start by calculating the time-averaged elongation rate $\bar{l}'_i$ for each cell $i$ in the wild-type data set, where the prime denotes a time derivative, by dividing the length added between birth and division by the total growth time. We then assume that the elongation rate for a cell at a time $t$ is approximately given by a rescaling of the population-averaged elongation rates $L'(t, l_b)$ by the time-averaged elongation rate of the cell $\bar{l}'_i$. Specifically, we calculate the estimated elongation rate at time $t$ by

$$l'_i(t) = L'(t, l_b) \frac{\bar{l}'_i n_i}{\sum_{t=0}^{t^i_{\mathrm{div}}} L'(t, l_b)}, \tag{A10}$$

with $n_i$ the number of time intervals in the growth trajectory of cell $i$, and $t^i_{\mathrm{div}}$ its division time. For times $t$ later than the first population division event $T_{\mathrm{div}}$, we obtain a value for $L'(t, l_b)$ by extrapolating the linear growth regime, setting $L'(t, l_b) = <L'(t, l_b)>_{20\,\mathrm{min}<t<T_{\mathrm{div}}}$.

From the ensemble $\{l'_i(t)\}$ of estimated elongation rates of all cells at each time since birth, we calculate the average elongation rate as a function of cell length by taking a moving average over the corresponding measured $\{l_i(t)\}$. The standard error on the mean is calculated from the standard deviation and the number of cells of each moving average bin.

# Appendix 9

## RAG model fitting procedure

The model fits shown in Main Text *Figure 6E–G* are obtained via the ParametricNDSolve function in Mathematica. The obtained parameter values are shown in *Appendix 9—tables 1* and *2*.

**Appendix 9—table 1.** Parameter values obtained by fitting Main Text *Equation (2)* to inferred elongation rate curves.
The values shown in column 4 and 6 are an average over the four birth lengths of each condition.

| Genotype | $l_b$ $[\mu m]$ | $\beta$ $[t^{-1}]$ | $\langle\beta\rangle$ $[t^{-1}]$ | $\frac{N(t=0)}{N^{max}}$ | $\left\langle\frac{N(t=0)}{N^{max}}\right\rangle$ |
|---|---|---|---|---|---|
| wild-type | 2.1 | 0.088 | 0.085 | 0.67 | 0.62 |
| | 2.3 | 0.068 | | 0.62 | |
| | 2.5 | 0.093 | | 0.62 | |
| | 2.7 | 0.089 | | 0.58 | |
| divIVA::divIVA-mCherry | 2.0 | 0.109 | 0.088 | 0.69 | 0.80 |
| | 2.2 | 0.100 | | 0.77 | |
| | 2.4 | 0.087 | | 0.84 | |
| | 2.6 | 0.054 | | 0.88 | |
| divIVA::divIVA-mCherry ΔrodA | 1.7 | 0.063 | 0.087 | 0.61 | 0.64 |
| | 1.9 | 0.094 | | 0.65 | |
| | 2.1 | 0.084 | | 0.64 | |
| | 2.3 | 0.11 | | 0.65 | |

**Appendix 9—table 2.** Parameter values obtained by fitting Main Text *Equation (3)* to inferred elongation rate curves.
The values shown in columns 5 and 7 are an average over the four birth lengths of each condition.

| Genotype | $l_b$ $[\mu m]$ | $\beta$ $[t^{-1}]$ | $\gamma$ $[t^{-1}]$ | $\langle\beta e^{\gamma t}\rangle_{t<20min}$ $[t^{-1}]$ | $\frac{N(t=0)}{N^{max}}$ | $\left\langle\frac{N(t=0)}{N^{max}}\right\rangle$ |
|---|---|---|---|---|---|---|
| wild-type | 2.1 | 0.016 | 0.162 | 0.13 | 0.72 | 0.67 |
| | 2.3 | 0.039 | 0.086 | | 0.67 | |
| | 2.5 | 0.058 | 0.080 | | 0.65 | |
| | 2.7 | 0.082 | 0.050 | | 0.62 | |
| divIVA::divIVA-mCherry | 2.0 | 0.072 | 0.06 | 0.14 | 0.71 | 0.82 |
| | 2.2 | 0.050 | 0.09 | | 0.79 | |
| | 2.4 | 0.025 | 0.14 | | 0.86 | |
| | 2.6 | 0.005 | 0.25 | | 0.92 | |
| divIVA::divIVA-mCherry ΔrodA | 1.7 | 0.023 | 0.094 | 0.12 | 0.67 | 0.68 |
| | 1.9 | 0.039 | 0.092 | | 0.69 | |
| | 2.1 | 0.064 | 0.050 | | 0.67 | |
| | 2.3 | 0.064 | 0.100 | | 0.68 | |

## Appendix 10

### Population simulation method

The goal of the population growth simulations is to obtain the distribution of cellular birth lengths assuming two different growth modes: asymptotically linear and exponential elongation. Both simulations extract all necessary growth parameters and distributions from the experimental data. For the asymptotically linear growth mode, the simulation serves as a check whether the assumed growth mode indeed recovers the correct cellular length distribution. For the exponential growth scenario, the simulation reveals the cellular length distribution an exponential grower would have if it had inherent noise levels similar to *C. glutamicum* allowing for a fair comparison. Both simulations start with a single cell and continue for 20 generations, after which the birth lengths of the last generation are binned and plotted. Repeated simulations with different lengths of the starting cell do not show discernable differences.

### Exponential growers

For the exponential growers, cells are assumed to elongate according to

$$l(t) = l_b \exp(\alpha t) + \zeta(t) \tag{A11}$$

The exponential growth rate $\alpha$ is chosen as the slope of the linear fit of $\ln\left(\frac{l_d}{l_b}\right)$ versus $t_d$ that intersects the origin, as shown in Main Text *Figure 3B*. A size-additive noise term is indicated by $\zeta(t)$, which will be specified below at the time of division. For a cell with a given birth length $l_b$, the target final length $l_t$ is determined via a linear fit of $l_b$ versus $l_d$, as shown in Main Text *Figure 3A*. The target growth time $t_t$ is then given by $t_t = \frac{1}{\alpha} \ln\left(\frac{l_t}{l_b}\right)$. A time additive noise term $\Delta t$ is added to $t_t$ according to experimentally observed growth time variations (*Appendix 10—figure 1D*). Additionally, a size-additive noise term $\Delta l$ encodes the division length variation due to $\zeta(t)$, which is also directly obtained from experiment (*Appendix 10—figure 1E*).

The full expression for the division length $l_d$ is then given by

$$l_d = l_b \exp(\alpha(t_t + \Delta t)) + \Delta l \tag{A12}$$

At division, the characteristic V-snap of *C. glutamicum* occurs, separating the two daughter cells. During this V-snap, the length of the daughter cells rapidly increases: the average measured birth length is 0.57 times the average measured division length (2.3 μm and 4.0 μm respectively), instead of the expected ratio of 0.5. To account for this V-snap effect, we calculate the distribution of added lengths during the V-snap. We find that the average added length depends on the division length: longer cells add less length during the V-snap than shorter cells (*Appendix 10—figure 1B*). To take this length dependence into account, we subdivide the data set into three division length bins, and obtain a distribution of added lengths during the V-snap for each bin. When a simulated cell divides, an added length during V-snap is randomly drawn from the distribution corresponding to its division length.

After division, the length asymmetry of the two daughter cells is chosen by drawing a random value from the experimentally observed division asymmetry distribution (*Appendix 10—figure 1C*) corresponding to the obtained division length. This distribution is found to be narrower for the shortest birth lengths (*Appendix 10—figure 1C*), thus two distributions are used.

### Asymptotically linear growers

For asymptotically linear growers, cells are assumed to elongate according to

$$l(t) = l_b + \lambda t + \gamma(\exp(-\beta t) - 1) + \eta(t), \tag{A13}$$

which is obtained by inserting Main Text *Equation 3* into Main Text *Equation 1*, integrating and grouping constant terms into $\lambda$ and $\gamma$. An additive noise term $\eta(t)$ is added to this to account for single-cell variability around the inferred average growth trajectory. We assume the cells to have the

same target final length $l_t$ as in the exponentially growing scenario, determined via a linear fit of $l_b$ versus $l_d$. For cells close to observed division times, the term proportional to $\gamma$ can be approximated as being constant in time, simplifying the growth mode to linear growth (Main Text *Figure 5A*). A time-additive noise term $\Delta t$ will then act as size-additive noise and can thus be absorbed into one additive noise term $\Delta l$, obtained from the experimental distribution of final sizes $l_d$ around the target final sizes (*Appendix 10—figure 1F*). The expression for the division length is thus given by

$$l_d = l_t + \Delta l. \tag{A14}$$

The division asymmetry and V-snap effect are incorporated in the same way as for the exponential grower simulation.

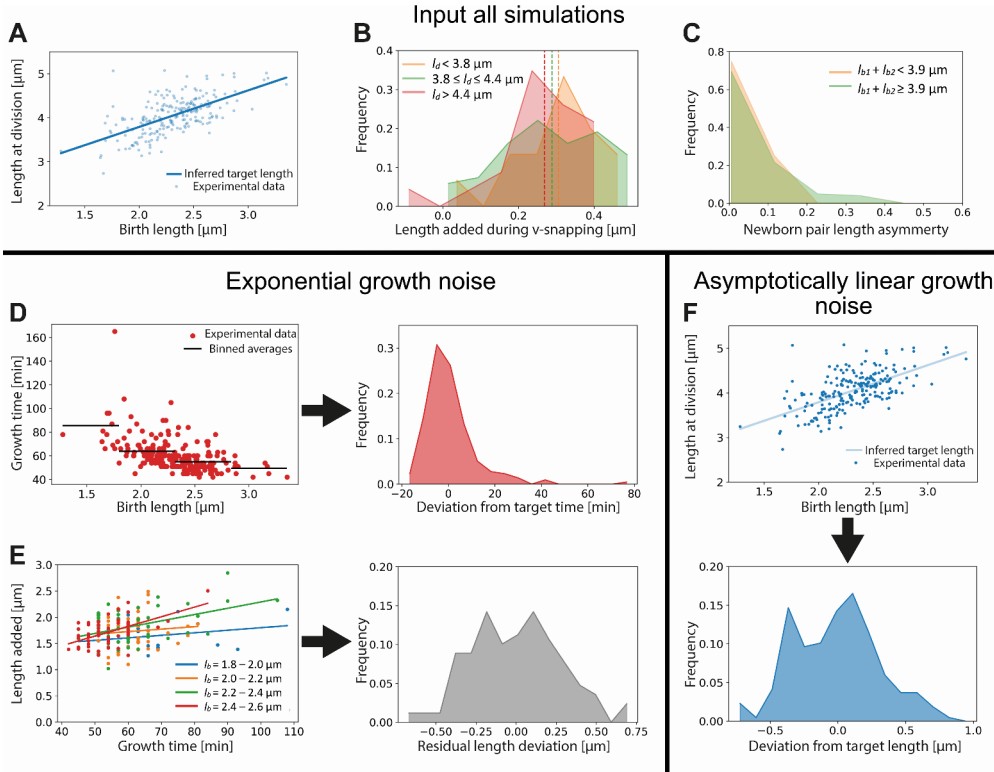

**Appendix 10—figure 1.** Input used for simulations of exponential and asymptotically linear growth. For both simulations, a linear fit of the division length versus birth length is used to define a target length (**A**). The length added during the V-snap at division is randomly drawn from the distribution corresponding to the division length of the simulated cell (**B**). The experimental data is divided into three subpopulations according to division length (red, green, and orange distributions), as the average length added during V-snap decreases with division length (dashed lines). The asymmetry of the daughter cells is randomly drawn from the distribution corresponding to the combined length of the simulated daughters (**C**). As the asymmetry is lower for the smallest daughter cells, the experimental data is divided into two subpopulations (red and green distributions). For the simulation of exponential growth, two noise sources are needed as input. The time-additive noise is randomly drawn from the distribution of deviations from target growth times (**D**). This distribution is obtained from the deviations of single-cell growth times from the average of their birth length bin. All growth variability not captured by growth time variations is calculated for four narrow birth length bins (blue, orange, green, and red points) (**E**). From the distribution of deviations of added lengths from a linear fit for each initial size bin, a size-additive noise term is randomly drawn. For the linear growth simulation, only a single additive noise term is required, which is randomly drawn from the distribution of deviations of cells lengths at division from the target division length (**F**).

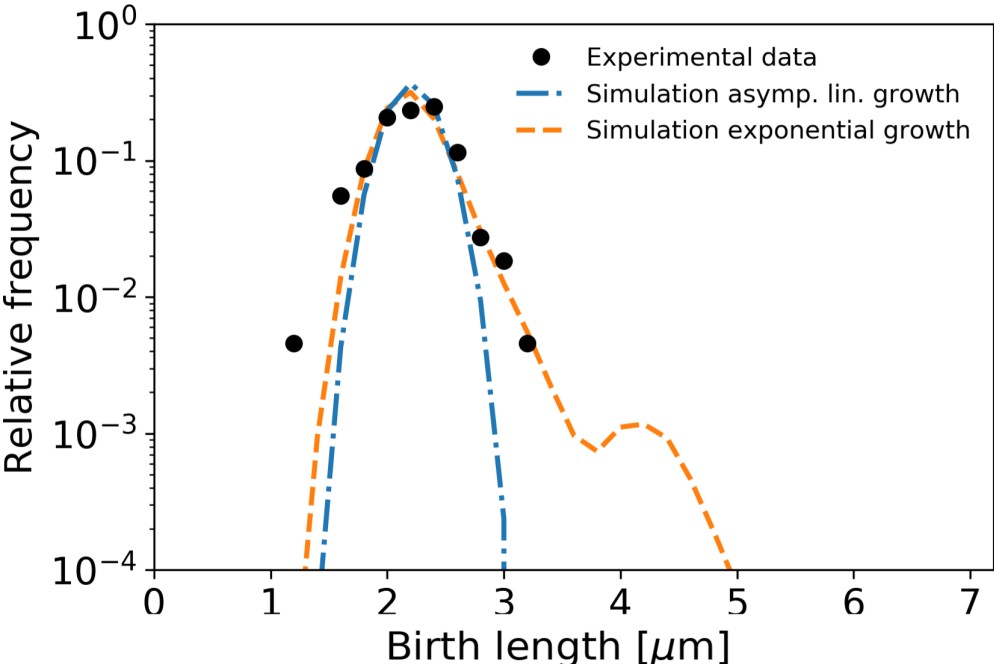

**Appendix 10—figure 2.** Birth length distributions as in Main Text *Figure 7*, but with single-cell variability in division symmetry, growth time, and (residual) length deviation reduced by a factor 3. The second peak in the length distribution of exponential growth is attributed to the large time deviation of one single cell seen in *Appendix 10—figure 1D*.

