## [Decision Letter]

**Acceptance summary:**

The paper presents a new and interesting method for inferring the growth dynamics of bacteria from noisy single-cell data and applies this to *C. glutamicum*. The method which is original, is promising for other systems.

**Decision letter after peer review:**

Thank you for submitting your article "Single-cell growth inference of *Corynebacterium glutamicum* reveals asymptotically linear growth" for consideration by *eLife*. Your article has been reviewed by 2 peer reviewers, and the evaluation has been overseen by a Reviewing Editor and Aleksandra Walczak as the Senior Editor. The reviewers have opted to remain anonymous.

Essential revisions:

The work is nicely executed and it should be published in *eLife*. Generally the method is potentially more notable than the result, so strengthening the method's validation is a good idea. There are two main remaining issues (please also see the reviewer comments below and address all of them):

1. The issue of whether seeing non-exponential growth is a notable result in itself as advertised in the paper, especially in light of recent work by Bruggeman's group (Biphasic Cell-Size and Growth-Rate Homeostasis by Single *Bacillus subtilis* Cells, Current Biology 2020) that shows non-exponential growth for the well-studied model bacterium *B. subtilis*. This example of non-exponential growth is not discussed at length in the paper under review.

2. Regarding the analysis method, it would be useful to spell out the central novelty and the method should be validated more appropriately. For example, in Bruggeman's paper cited above they used a different analysis method (growth rate vs. age plot) that seems to work well, and the two methods were not compared or discussed.

*Reviewer #1 (Recommendations for the authors):*

This is an interesting study, with a strong method, but also nice results for the specific system. Here are a few points where the authors could improve the manuscript:

Main issues:

– Change the text to remove the repeated references to the consensus of exponential growth, this is likely less dominant than claimed here. E.g. abstract, on p. 2 bottom, p.4 line 80 ("universality") etc.

– Add some analysis to test whether linear growth continues into the phase where the first cells have started to divide. I imagine that removing the earliest dividing cells before the elongation rate determination should give the opportunity to analyze the remaining ones for a longer time. This could be done successively, and while the growth rate would be reduced as more and more cells are removed, one should see that growth rate remains constant up to later times.

*Reviewer #2 (Recommendations for the authors):*

Some comments:

1. The trend of elongation rate at birth seems to be different for different mutants. Figure 5A shows that in *C. glutamicum* divIVA:divIVA-mCherry the elongation rate is higher for cells with longer size at birth. Can the authors comment on this and how this is to be interpreted within their model?

2. Similarly, could the authors explain why the asymptotic linear elongation rate is smaller in *C. glutamicum* divIVA:divIVA-mCherry such cells as compared to WT using the RAG model?

3. In Figure 7, the authors seem to compare the birth length distribution obtained from theory and experiment. However, the fitness advantage or disadvantage of a longer-tailed distribution is unclear. In my opinion, the section can be removed from the main text for a better reading of the paper.

[Editors' note: What follows are minor comments after acceptance.]

*Reviewer #2 (Recommendations for the authors):*

The authors have done a good job of addressing the comments and I find the paper adequate for publication in eLife. I would like to raise a potential issue with the agreement of the model and data implicated by Author response image 1 of the response letter: In the biorxiv paper referenced in the updated version (by Kar et al.), a plot of the form of Author response image 1 is shown to be linear (y=x) both for exponential and linear growth. It therefore seems that for the model of growth proposed in the work (exponential crossing to linear) one would also obtain a simple y=x dependence. However Author response image 1 shows significant deviations from such dependence, suggesting that the proposed model might be inconsistent with the data?

---

## [Author Response]

Essential revisions:The work is nicely executed and it should be published in eLife. Generally the method is potentially more notable than the result, so strengthening the method's validation is a good idea. There are two main remaining issues (please also see the reviewer comments below and address all of them):1. The issue of whether seeing non-exponential growth is a notable result in itself as advertised in the paper, especially in light of recent work by Bruggeman's group (Biphasic Cell-Size and Growth-Rate Homeostasis by Single *Bacillus subtilis* Cells, Current Biology 2020) that shows non-exponential growth for the well-studied model bacterium *B. subtilis*. This example of non-exponential growth is not discussed at length in the paper under review.

We thank the reviewers for pointing out these recent developments. Whereas cell-cycle averaged measures of growth have revealed exponential single cell growth for a range of bacterial species over the last decade, detailed inspection of growth rate and elongation rate have revealed deviations from exponential growth over the last year. Using this method, such deviations have been found recently in (Nordholt et al., Current Biology 2020), but also in a preprint that appeared on the BioRxiv about a week ago in (To bin or not to bin: analyzing single-cell growth data, Kar et al., BioRxiv, 2021). In light of these results, we have modified our introduction and our discussion of exponential growth. We made the following changes to the Main Text:

Page 2, lines 23-24:

“So far, single bacterial cells have been found to grow predominantly exponentially, which implies the need for tight regulation to maintain cell size homeostasis.”

Page 2, lines 42-45:

“Extensive work was done on common model organisms, including *Escherichia coli*, *Bacillus subtilis*, and *Caulobacter crescentus*, revealing that averaged over the cell cycle, single cells grow exponentially for these species(Taheri-Araghi et al. 2015; Mir et al. 2011; Iyer-Biswas et al. 2014; Yu et al. 2017; Godin et al. 2010).”

Page 3, lines 50-55:

“Recently, detailed analysis of the mean growth rate throughout the cell cycle revealed deviations from pure exponential growth. For *B. subtilis* (Nordholt, van Heerden, and Bruggeman 2020), a biphasic growth mode was observed, where a phase of approximately constant elongation rate is followed by a phase of increasing elongation rate. For *E. coli*, a new method provides evidence for super-exponential in the later stages of the cell cycle (Kar et al. 2021) A promising candidate for uncovering strong deviations from exponential growth is the Grampositive Corynebacterium glutamicum.”

Page 27, lines 558-560:

“The inferred asymptotically linear growth of *C. glutamicum* deviates from the predominantly found exponential single-cell bacterial growth, and suggests the presence of novel growth regulatory mechanisms.”

2. Regarding the analysis method, it would be useful to spell out the central novelty and the method should be validated more appropriately. For example, in Bruggeman's paper cited above they used a different analysis method (growth rate vs. age plot) that seems to work well, and the two methods were not compared or discussed.

We have now extended our Discussion section to include a comparison of our method to that of (Nordholt et al., Current Biology 2020), and of (Kar et al., BioRxiv, 2021). The paragraph in which we discuss this has been revised to (page 25, lines 501-511):

“An analysis of elongation rates as a function of time and birth length has previously been done in *B. subtilis* by binning cells based on birth length (Nordholt, van Heerden, and Bruggeman 2020). Applying this method to our data set yields elongation rates averaged over cells within a binning interval (Appendix 2—figure 5). Averaging our inferred elongation rates over the same bins, we find the two methods to yield consistent results. The binning method, however, involves a tradeoff: a smaller bin width results in a larger error on the inferred elongation rates, whereas a larger bin width averages out all variation within a larger birth length interval. Our method does not suffer from this binning-related tradeoff, and it provides detailed elongation rate curves at any given birth length. In other recent work (Kar et al., 2021), average growth rate curves were calculated as a function of cell phase. Our method provides additional detail by extracting the dependence of elongation rate on birth length as well as time since birth.”

In (Kar et al., bioRxiv, 2021), it was further shown that the test used for exponential growth shown in Figure 3B is unreliable in the presence of noise in the exponential growth rate. We made the following change to discuss this point (page 11, lines 190-192):

“Furthermore, it was recently shown that exponentially growing cells can appear nonexponential with this test in the presence of noise in the exponential growth rate (Kar et al. 2021).”

This shortcoming of this exponential growth test does not impact subsequent results, since this test is only used as a motivation to study elongation rates in more detail. In (Kar et al., bioRxiv, 2021), an alternative test is proposed which bins over the ln⁡(ldlb) axis rather than the time axis, which is reported to not be subject to this bias in the presence of noise. Performing this test on our data still shows a deviation from exponential growth (Author response image 1).

**Author response image 1. respfig1:** ln⁡(ldlb) versus generation time for all cells (blue dots) and the average per generation time bin (orange squares), with the standard error of the mean shown for all generation times for which at least 3 data points are available.

Reviewer #1 (Recommendations for the authors):This is an interesting study, with a strong method, but also nice results for the specific system. Here are a few points where the authors could improve the manuscript:

We appreciate these supportive comments and constructive suggestions.

Main issues:– Change the text to remove the repeated references to the consensus of exponential growth, this is likely less dominant than claimed here. E.g. abstract, on p. 2 bottom, p.4 line 80 ("universality") etc.

As mentioned in our response to Essential Revision 1, we modified our discussion of exponential growth to include these developments in our revised manuscript.

– Add some analysis to test whether linear growth continues into the phase where the first cells have started to divide. I imagine that removing the earliest dividing cells before the elongation rate determination should give the opportunity to analyze the remaining ones for a longer time. This could be done successively, and while the growth rate would be reduced as more and more cells are removed, one should see that growth rate remains constant up to later times.

In the supplement we had included one analysis, in which we computed conditional elongation curves, conditioned on cells that grew for at least 51 minutes. Following the reviewer’s suggestion, we now expanded this analysis to show mean elongation curves for a range of choices for the cutoff time. The results are shown in the new Appendix 2—figure 4. We find the resulting curves to be consistent with linear growth. However, the error on the inferred mean elongation rates increases with higher cutoff times due to fewer cells being included in the analysis.

To provide stronger evidence for the continuation of linear growth until division, we adapted our inference procedure to describe mean elongation curves as a function of the time until division and division length. Using this procedure, we indeed find that linear growth continues until division. We explain this procedure as follows in the Main Text, page 15, lines 289-296:

“To further test if the linear growth mode persists until division, we adapt our inference procedure to obtain average elongation curves L(t−td, ld) as a function of the time until division t−td and division length ld. The construction is analogous to that of L(t,lb) (Appendix 6). Calculating the corresponding elongation rate curves, we find that linear growth indeed extends until the division time across division lengths (inset Figure 5B, SI Appendix 6—figure 1). Note that with this construction, elongation rates become biased once |t−td| exceeds the shortest single-cell total growth time. Hence, for our analysis we only consider elongation rate curves until this point.”

The results are shown in the new inset of Figure 5 and in the new SI Appendix 6.

Reviewer #2 (Recommendations for the authors):Some comments:1. The trend of elongation rate at birth seems to be different for different mutants. Figure 5A shows that in C. glutamicum divIVA:divIVA-mCherry the elongation rate is higher for cells with longer size at birth. Can the authors comment on this and how this is to be interpreted within their model?

The inferred mean elongation rates indeed suggest a change in the dependence of elongation rate at birth on birth length. To see how this could be interpreted within our model, we first note that at the end of the cell cycle, cells with a larger birth length grow faster on average. As the old poles are inherited by the daughter cells, we would thus expect cells with a larger birth length to have a faster growing old pole. To still have similar elongation rates across birth lengths, this would imply that the new pole is on average less saturated at birth for longer cells, and more saturated at birth for shorter cells.

It is, however, unclear if such differences in birth elongation rates between our strains are indeed present. The error margins on the elongation rates at birth are significant: A spread of mean elongation rates over birth lengths similar to that of the DivIVA labelled cells would still be within the error margins for both the wild type and the ΔrodΔ strains. Because of this, we decided not to discuss this point in our manuscript.

2. Similarly, could the authors explain why the asymptotic linear elongation rate is smaller in C. glutamicum divIVA:divIVA-mCherry such cells as compared to WT using the RAG model?

Within the RAG model, there could be several possible explanations for the lowering of the elongation rates in the DivIVA labelled cells compared to wild type:

– An overall decrease of the amount of transglycosylase sites N(t) at the cell tip

– A lowering of the insertion efficiency of the transglycosylases α

– A lowering of the local concentration of lipid-II C(t) at the tip.

Without further experiments to elucidate how the DivIVA labelling affects the apical growth mechanism, it is not possible to narrow this down further. Therefore, we decided not to discuss this labelling in the context of the RAG model in the present manuscript. This result is consistent, however, with earlier findings of an overall decrease of the average single-cell elongation rate for DivIVA labelled cells. To emphasize this, we added the following to the Main Text on page 15, lines 285-288:

“This likely reflects a disturbance in the interaction between RodA or bifunctional PBPs and the DivIVA-mCherry fusion protein, indicating that the DivIVA-mCherry fusion is not fully functional. This is consistent with findings we reported earlier (Donovan et al., PLOS One 2013)”

3. In Figure 7, the authors seem to compare the birth length distribution obtained from theory and experiment. However, the fitness advantage or disadvantage of a longer-tailed distribution is unclear. In my opinion, the section can be removed from the main text for a better reading of the paper.

While it is difficult to determine with certainty which fitness advantages this provides for *C. glutamicum*, we find it an interesting observation how the growth mode affects this distribution (if all other parameters remain the same). The main point we intended to make with this figure, is that linear growth can act as a cell-size regulation mechanism.

To emphasize this point, we made the following addition in the Main Text on page 24, lines 478-481:

“*C. glutamicum* has a high degree of variation of division symmetry (Appendix 9—figure 1C) and single-cell growth times, but due to the asymptotically linear growth mode, the population-level variations in cell size are still relatively small. This indicates that linear growth can act as a regulator for cell size.”

[Editors' note: What follows are minor comments after acceptance.]Reviewer #2 (Recommendations for the authors):The authors have done a good job of addressing the comments and I find the paper adequate for publication in eLife. I would like to raise a potential issue with the agreement of the model and data implicated by Author response image 1 of the response letter: In the biorxiv paper referenced in the updated version (by Kar et al.), a plot of the form of Author response image 1 is shown to be linear (y=x) both for exponential and linear growth. It therefore seems that for the model of growth proposed in the work (exponential crossing to linear) one would also obtain a simple y=x dependence. However Author response image 1 shows significant deviations from such dependence, suggesting that the proposed model might be inconsistent with the data?

Indeed in Kar et al, it is shown that a plot of the form of Author response image 1 obeys y=x both for exponential growth and linear growth within an adder model. This, however, does not imply that a model that switches between these growth modes during growth also obeys the same y=x trend. To illustrate this, we performed simulations of cells growing purely exponentially, purely linearly, and switching from exponential to linear growth at a fixed moment in the cell cycle (Author response image 2). We find that purely exponentially growing cells and purely linearly growing cells obey the y=x trend, in agreement with Kar et al. By contrast, simulated cells performing mixed growth deviate from this trend, indicating that mixed models, as we show to be the case for Corynebacterium, in general do not obey the y=x trend.

**Author response image 2. respfig2:** ln⁡(ldlb) versus generation time for three simulated datasets (blue dots), and the average per generation time (orange squares), with the standard error of the mean. The orange line represents a linear fit through the generation time averages that passes through the origin. For each simulation, birth length and division length pairs were sampled from the DivIVA-labelled cell dataset. The number of such pairs is equal to the number of cells in the data set. Each simulated cell started at an assigned birth length, and was grown until the minimum difference with the corresponding division length was reached. The inferred measurement noise (Appendix 3) was applied to all simulated data points. (**A**) Exponential growth, constructed by setting l(t)=lbe0.26t. (**B**) Linear growth, constructed by setting 𝑙(𝑡) = 𝑙_b_ + 2.08𝑡. (**C**) A mixed model, where cells grow exponentially for the first 30 minutes, and switch to linear growth from then on.